# Protein analysis of extracellular vesicles to monitor and predict therapeutic response in metastatic breast cancer

Fei Tian[1,2,5], Shaohua Zhang [3,5], Chao Liu [1,2✉], Ziwei Han[1,2], Yuan Liu[1], Jinqi Deng[1], Yike Li[1,2], Xia Wu[3], Lili Cai[4], Lili Qin[3], Qinghua Chen[1,2], Yang Yuan[3], Yi Liu[3], Yulong Cong[4], Baoquan Ding [1,2], Zefei Jiang [3✉] & Jiashu Sun [1,2✉]

Molecular profiling of circulating extracellular vesicles (EVs) provides a promising non-invasive means to diagnose, monitor, and predict the course of metastatic breast cancer (MBC). However, the analysis of EV protein markers has been confounded by the presence of soluble protein counterparts in peripheral blood. Here we use a rapid, sensitive, and low-cost thermophoretic aptasensor (TAS) to profile cancer-associated protein profiles of plasma EVs without the interference of soluble proteins. We show that the EV signature (a weighted sum of eight EV protein markers) has a high accuracy (91.1 %) for discrimination of MBC, non-metastatic breast cancer (NMBC), and healthy donors (HD). For MBC patients undergoing therapies, the EV signature can accurately monitor the treatment response across the training, validation, and prospective cohorts, and serve as an independent prognostic factor for progression free survival in MBC patients. Together, this work highlights the potential clinical utility of EVs in management of MBC.

---

[1] Beijing Engineering Research Center for BioNanotechnology, CAS Key Laboratory of Standardization and Measurement for Nanotechnology, CAS Center for Excellence in Nanoscience, National Center for Nanoscience and Technology, Beijing, China. [2] School of Future Technology, University of Chinese Academy of Sciences, Beijing, China. [3] Department of Breast Cancer, The Fifth Medical Centre, Chinese PLA General Hospital, Beijing, China. [4] Department of Laboratory Medicine, The Second Medical Center, Chinese PLA General Hospital, Beijing, China. [5] These authors contributed equally: Fei Tian, Shaohua Zhang. ✉email: liuc@nanoctr.cn; jiangzefei@csco.org.cn; sunjs@nanoctr.cn

Metastatic breast cancer (MBC) is a heterogeneous disease comprising multiple distinct subtypes and remains one of the primary causes of cancer death among females worldwide[1]. Despite of the increasing availability of diagnosis and treatment, 25–28% of breast cancer (BC) patients are diagnosed as MBC with a 5-year survival rate of 27%[2]. In patients with MBC, real-time monitoring and prediction of therapy responses is critical to optimal personalized treatment regimens[3]. Although tissue biopsy has been used for diagnosis of MBC, its invasiveness introduces risk and morbidity, and patients may not have adequate tissue available at the time of progression[4]. Serial imaging by computed tomography (CT) suffers from a reduced sensitivity in monitoring treatment response and cannot be used for the prediction of disease progression[5]. Thus, there is an urgent need of reliable, noninvasive tools for diagnosis, monitoring, and prognosis of MBC.

Blood test offers an attractive alternative for MBC management by analyzing circulating tumor-related biomarkers in a minimally invasive and repeatable manner[6]. Carcinoma antigen 15-3 (CA 15-3) is the most widely used plasma/serum biomarker in monitoring MBC, yet its response parallels disease response in only half of the patients[7,8]. The quantification and genetic characterization of circulating tumor cells (CTCs) and circulating tumor DNA (ctDNA) have good concordance with radiographic measurement in monitoring tumor burden and are predictive of disease progression and survival for MBC patients[8–10]. However, owing to the low abundance of CTCs and ctDNA in peripheral blood, the analysis of CTCs and ctDNA often requires large sampling volumes and sophisticated methodologies to achieve satisfied sensitivity[11].

Tumor-derived extracellular vesicles (EVs) have recently emerged as an important class of circulating biomarkers for cancer diagnosis[12–18]. They are nano/micrometer-sized, lipid bilayer-enclosed vesicles that contain a multitude of biological molecules (proteins, nucleic acids, lipids, etc) from parental tumor cells. A single tumor cell can release more than $10^4$ EVs per day, making tumor-derived EVs more abundant than other circulating biomarkers[19,20]. EV-bound proteins have been demonstrated to play important roles in critical processes for BC progression and metastasis, including tumor vascularization[21,22], HER2-targeted therapy resistance[23], matrix remodeling[24], immune evasion[25], and premetastatic niche formation[26,27]. Therefore, serial sampling of EV protein markers from blood may facilitate the diagnosis and monitoring of MBC, yet remains to be discovered in clinical cohort studies.

Current analysis of EVs is restricted by the time-consuming ultracentrifugation for EV isolation and the lack of ultrasensitive and specific assay for EV detection without the interference from non-vesicular contaminations[28–30]. To directly address the aforementioned challenges, we have previously developed a straightforward, sensitive, and cost-effective thermophoretic aptasensor (TAS) to determine the surface protein profiles of serum EVs from cancer patients, without the need for EV pre-isolation[31]. The operating principle of TAS relies on thermophoretic enrichment (>$10^3$-fold) of aptamer-bound EVs in a size-dependent manner to generate an amplified fluorescence signal, whose intensity is indicative of the expression level of EV surface protein. Applying a machine-learning algorithm, the EV signature of seven protein markers is defined for early detection and classification of six different cancer types.

Here, we set out to explore the utility of EV protein markers for MBC diagnosis, treatment response monitoring, and prognostic prediction using the TAS platform. A machine-learning algorithm is devised to identity the EV signature on the basis of expression levels of eight BC-associated EV protein markers. The EV signature offers high accuracy to discriminate MBC from non-metastatic breast cancer (NMBC) and healthy donors (HD), and to monitor MBC treatment response in training, validation, and prospective cohorts. The EV signature is also associated with progression-free survival (PFS) of MBC patients undergoing therapies. Notably, we discover that EV PSMA, a generally recognized prostate cancer biomarker, is a significant biomarker for monitoring and prognostic discrimination of MBC patients.

## Results

**Thermophoretic aptasensor for sensitive analyses of EV protein markers.** A number of putative BC-associated protein markers were selected to be detected on EVs. CA 15-3 is the most widely used marker in evaluating therapy response and disease recurrence in BC patients. CA 125 and carcinoembryonic antigen (CEA) have also been used to monitor recurrence and metastasis of BC[32,33]. Human epidermal growth factor receptor 2 (HER2) and epidermal growth factor receptor (EGFR) are ErbB family members, which are frequently overexpressed in BC and serve as therapeutic targets[34,35]. Prostate-specific membrane antigen (PSMA) is associated with the neovasculature in primary and metastatic tumor sites in BC[36,37]. Epithelial cell adhesion molecule (EpCAM) is overexpressed in various cancer types including BC[38]. Vascular endothelial growth factor (VEGF) promotes tumor angiogenesis and lymphangiogenesis in the progression and metastasis of BC[39]. The signaling, functions, and pathways of the eight protein markers have been depicted in Supplementary Table 1.

The TAS platform was developed for analyses of proteins specifically present on EVs with clinical feasibility (Fig. 1). Clinical plasma sample (1 μL, diluted by 100-folds) was incubated with Cy5-conjugated aptamers for 2 h to enable the binding of aptamers to the target EV proteins (Supplementary Tables 2 and 3). The use of aptamers rather than antibodies for protein detection has the benefits of higher thermal stability and cost-effective production. The incubated sample was then subjected to a 10-min localized laser heating for size-dependent thermophoretic accumulation of EVs (~100 nm in mode size) to amplify the fluorescence signal of aptamer-bound EVs (Fig. 1a–c and Supplementary Fig. 1). Importantly, soluble proteins of small sizes (a few nm in size) could not be accumulated and detected by TAS due to their weak thermophoresis[40]. In TAS experiments, the fluorescence signal from EV-depleted plasma (sample i) was weak and similar to that in EV-depleted plasma spiked with soluble proteins such as CA 15-3 (25 U mL$^{-1}$), CA 125 (35 U mL$^{-1}$), or CEA (5 ng mL$^{-1}$) (sample ii). In contrast, a fourfold increase in fluorescence intensity was observed in sample iii, which was prepared by spiking plasma EVs from an untreated MBC patient ($2 \times 10^9$ mL$^{-1}$) into sample ii (Fig. 1d). These results suggested that TAS allowed the detection of EV protein markers without the interference of soluble counterparts. The limit of detection (LoD, 3× standard deviation above the blank) by TAS was $3.8 \times 10^7$ mL$^{-1}$ for EVs, which was >$10^2$-fold more sensitive than ELISA (Fig. 1e). Given the high concentration of EVs in plasma ($10^{10}$–$10^{11}$ mL$^{-1}$)[41], TAS could be performed directly on a small plasma sample volume (<1 μL).

We first characterized the performance of TAS by measuring the expression levels of the eight protein markers on EVs derived from different BC cell lines (BT-474, SK-BR-3, and MDA-MB-231) and benign mammary epithelial cell line (MCF-10A). The number of EVs from different cell lines was quantified by NTA and the same number of EVs ($10^{10}$ mL$^{-1}$) was implemented in the assay (Fig. 1f, g). Experimental results showed that HER2 was overexpressed on EVs from SK-BR-3 and BT-474, while EGFR and VEGF were overexpressed on EVs from MDA-MB-231, which were in good agreement with previous studies[42,43]. Moreover, the expression of eight individual EV protein markers

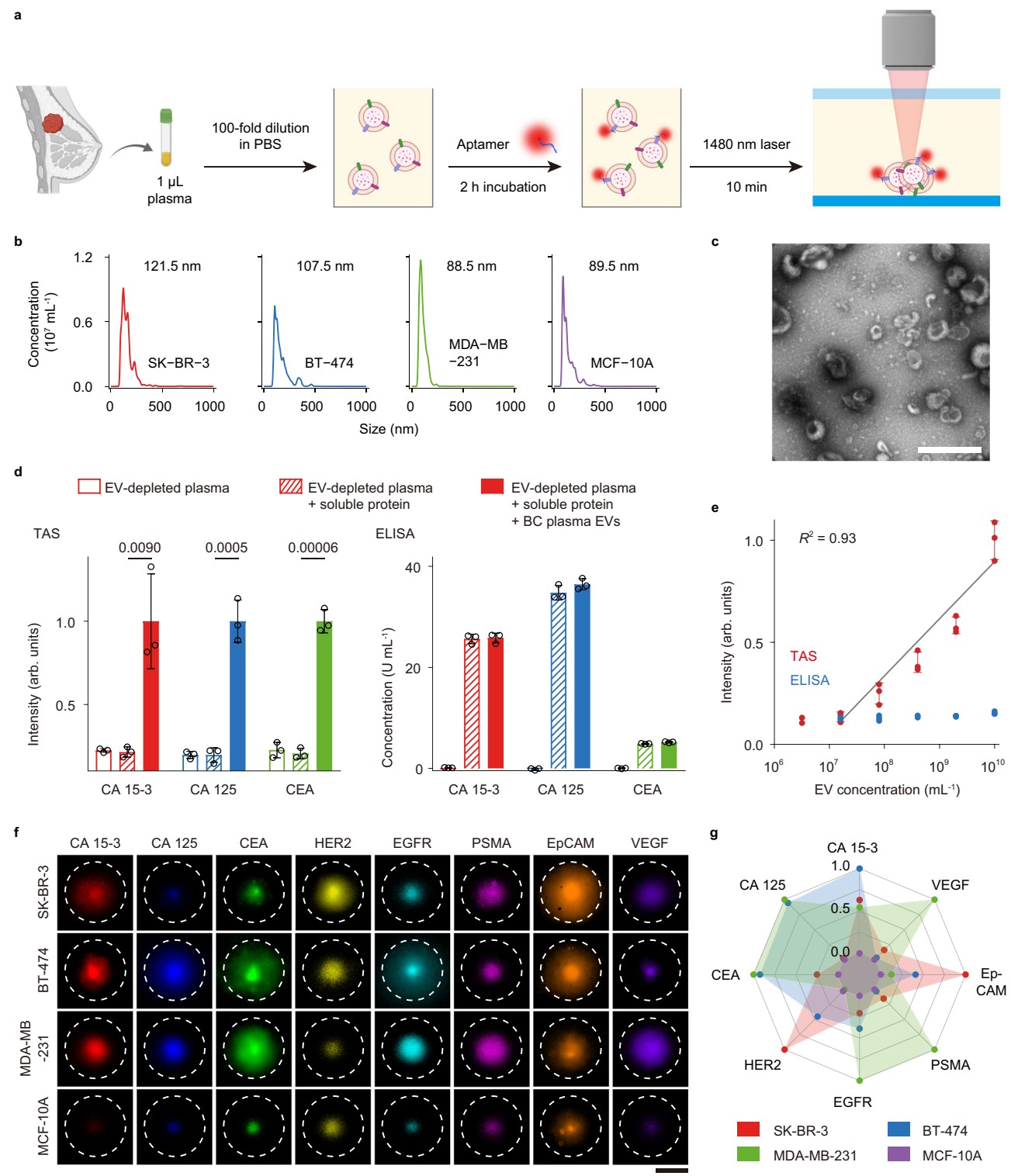

from BC cell lines was higher compared to MCF-10A. Therefore, we decided to assay these eight protein markers on plasma EVs in clinical samples.

**EV protein profiles of BC patients.** For EV-based cancer diagnosis, 123 plasma samples were collected from 36 MBC patients before salvage treatment, 21 NMBC patients before surgical therapy, and 66 age-matched HD. The expression patterns of the eight protein markers on plasma EVs were detected by TAS

(Supplementary Table 4 and Supplementary Data). Fluorescence images of aptamer-labeled EVs after thermophoretic accumulation reflected elevated levels of eight EV markers from MBC or NMBC patients compared to HD (Fig. 2a, b). Both MBC and NMBC patients showed considerable heterogeneity in EV protein marker expression (Fig. 2c and Supplementary Fig. 2). Precision-Recall Curves (PRC) showed that CA 15-3 and EpCAM on EVs achieved a high discriminative capacity to differentiate BC from HD (area under the PRC (AUPRC) = 0.9286 for CA 15-3; AUPRC = 0.9709 for EpCAM; Fig. 2d, e and Supplementary

**Fig. 1 Thermophoretic aptasensor (TAS) for detecting protein markers of EVs. a** Schematic of the TAS procedure. Clinical plasma samples (1 µL, diluted by 100-folds) were incubated with Cy5-conjugated detecting aptamers to bind to target proteins on EVs, and then subjected to thermophoretic accumulation to amplify the fluorescence signal of aptamer-bound EVs, enabling rapid and sensitive detection of EV protein markers. **b** Size distribution of EVs derived from three BC cell lines SK-BR-3 (red line), BT-474 (blue line), and MDA-MB-231 (green line), and benign mammary epithelial cell line (MCF-10A, violet line) using nanoparticle tracking analysis (NTA). Size modes are indicated. **c** Wide-field TEM image of EVs. The representative image is shown from three independent repeats. Scale bar, 500 nm. **d** TAS and ELISA measurement of expression levels of CA 15-3 (red bars), CA 125 (blue bars), and CEA (green bars) in three types of samples: (i) EV-depleted plasma diluted by 100-folds in 1× PBS; (ii) the diluted EV-depleted plasma spiked with soluble proteins; (iii) the sample ii spiked with plasma EVs ($2 \times 10^9$ mL$^{-1}$, $n = 3$ samples for each protein marker). Statistical difference was determined by a two-sided, parametric $t$ test. $P$ value is indicated in the chart. **e** Sensitivity of TAS (red dots) and ELISA (blue dots) for the detection of plasma EVs incubated with CEA aptamer (0.1 µM) ($n = 3$ samples for each EV concentration). R square ($R^2$) is indicated. **f** Fluorescence images of aptamer-labeled EVs ($10^{10}$ mL$^{-1}$) after thermophoretic accumulation showing elevated levels of eight EV protein markers from the three BC cell lines, SK-BR-3, BT-474, and MDA-MB-231, compared to MCF-10A. Scale bar, 50 µm. Images are shown from a single measurement. **g** Radar plot showing TAS analyses of 8 EV protein markers from the four different cell lines (BT-474 is represented by red dots, SK-BR-3 by blue dots, MDA-MB-231 by green dots, and MCF-10A by violet dots). Error bars represent the mean ± s.d. in (**d**, **e**). Source data are provided as a Source Data file.

Table 5). However, no single marker had sufficiently high sensitivity and specificity in discriminating MBC and NMBC (AUPRC = 0.6704–0.9068) (Supplementary Table 5). We also tested the performance of clinical gold standard plasma biomarker (CA 15-3) for BC. Only 58.3% of MBC patients (21 of 36) and 14.3% of NMBC patients (3 of 21) showed increased levels of plasma CA 15-3 (>25 U mL$^{-1}$, threshold value in clinics). Moreover, our data showed weak correlations between any pair of EV protein markers (median Pearson correlation coefficient $r = 0.31$, Fig. 2f), and between plasma CA 15-3 and CA 15-3 on EVs ($r = 0.29$, Fig. 2f), which encouraged us to combine EV markers for accurate diagnosis of MBC.

**Establishing the EV$^{DX}$ signature for MBC diagnosis**. To improve the performance of EVs in differentiating MBC, NMBC, and HD groups, we harnessed machine-learning methods to compile all EV protein marker profiles. The EV$^{DX}$ signature, representing a weighted sum of CA 15-3, CA 125, CEA, HER2, EGFR, PSMA, EpCAM, and VEGF signals identified by linear discriminant analysis (LDA, Supplementary Software), had remarkably high accuracy (AUPRC = 0.9912) to distinguish BC from HD, and an AUPRC of 0.9433 for MBC and NMBC classification (Fig. 3a, b and Supplementary Table 5). The EV$^{DX}$ signature showed an overall accuracy of 91.1% across three classes (Fig. 3c). In contrast, the unweighted sum (SUM) of eight EV protein markers had a lower overall accuracy of 79.7% (95% CI = 71.5–86.4%) compared with the EV$^{DX}$ signature (Supplementary Tables 6–8). Although there was heterogeneity within the eight EV markers and the EV$^{DX}$ signature, segregation according to metastasis sites of MBC patients was not observed by hierarchical clustering analysis (Fig. 3d). However, expression of EGFR was higher on EVs from MBC patients with lung metastasis (Fig. 3e). Similarly, previous studies have observed that overexpression of EGFR at the cellular level was highly associated with lung metastasis in MBC patients[44,45]. Moreover, the EV signature against tumor size (EV$^{TS}$, the weighted sum of eight makers using multivariate linear regression) showed a good correlation with the sum of tumor sizes from 3 to 5 largest measurable lesions in MBC patients (mean square errors (MSE) = 19.3, $R$ squared ($R^2$) = 0.7954, $n = 20$ MBC patients, Fig. 3f and Supplementary Fig. 3). In NMBC patients, the EV$^{TS}$ signature was also correlated with the primary tumor size (MSE = 2.62, $R^2 = 0.9876$, $n = 11$ NMBC patients, Supplementary Fig. 4).

**The EV$^M$ signature for monitoring treatment response in MBC**. Next, we assessed the ability of EV protein profiles for monitoring treatment response in a cohort of 112 plasma samples collected from MBC patients after 1–4 periods of treatments

(Supplementary Tables 9 and 10). To construct the EV$^M$ signature for monitoring treatment response, 60% of plasma samples were randomly assigned to the training cohort. Fig. 4a summarized the relative changes in expression levels of EV protein markers (ΔIntensity) for patients with partial response (PR, $n = 18$), stable disease (SD, $n = 17$), and progressive disease (PD, $n = 10$) by Response Evaluation Criteria in Solid Tumors (RECIST, version 1.1)[46]. The EV$^M$ signature defined as the weighted sum of ΔIntensity of eight markers by LDA showed an area under the curve (AUC) of 0.9429 (95% confidential interval (95% CI) = 0.8711–1.0000) and an accuracy of 88.9% (95% CI = 76.0–96.3%) in differentiating PD from PR/SD by receiver operating characteristic (ROC) analysis (Fig. 4b, c). In the validation cohort including the remained 40% plasma samples from MBC patient, the EV$^M$ signature achieved an AUC of 0.9066 (95% CI = 0.7894–1.0000) and an accuracy of 87.9% (95% CI = 71.8–96.6%) in differentiating PD ($n = 7$) from PR/SD ($n = 26$) (Fig. 4b, d).

In a prospective cohort of 35 plasma samples, the EV$^M$ signature generated using the trained LDA model achieved an accuracy of 85.2% (95% CI = 66.3–95.8%) for discrimination between PD ($n = 7$) and PR/SD ($n = 20$) (Fig. 5a–c and Supplementary Tables 11 and 12). Across the training, validation, and prospective cohorts, the EV$^M$ signature showed similar performance for classifying treatment response when applied to different BC subtypes (AUC = 0.9444, 95% CI = 0.8681–1.0000 for hormone receptor-positive (HR +); AUC = 0.8674, 95% CI = 0.7307–1.0000 for human epidermal growth factor receptor 2-positive (HER2 +); AUC = 0.9026, 95% CI = 0.7662–1.0000 for triple-negative breast cancer (TNBC); Fig. 5d, e). Notably, although plasma CA 15-3 is currently the best plasma marker to follow treatment response in MBC patients, its discriminative capacity in PD versus PR/SD (AUC = 0.7903, 95% CI = 0.6675–0.9130) was inferior to the best EV protein marker, PSMA (AUC = 0.8447, 95% CI = 0.7335– 0.9560) (Supplementary Table 10). Despite of being a promising target in prostate cancer, PSMA expression was also observed in endothelial cells of tumor-associated neovasculature[47], and positively correlated with tumor invasiveness and progression in BC[36,37].

**The EV$^M$ signature for longitudinal monitoring in MBC**. In longitudinal studies, we compared the performance of EV$^M$ signature and plasma CA 15-3 in monitoring response to systemic treatments in MBC patients with at least three treatment points (Fig. 6 and Supplementary Fig. 5). The change in tumor burden could be better captured by the EV$^M$ signature than plasma CA 15-3 across different BC subtypes (Fig. 6 and Supplementary Fig. 5). Representatively, a HR + MBC patient with continuous

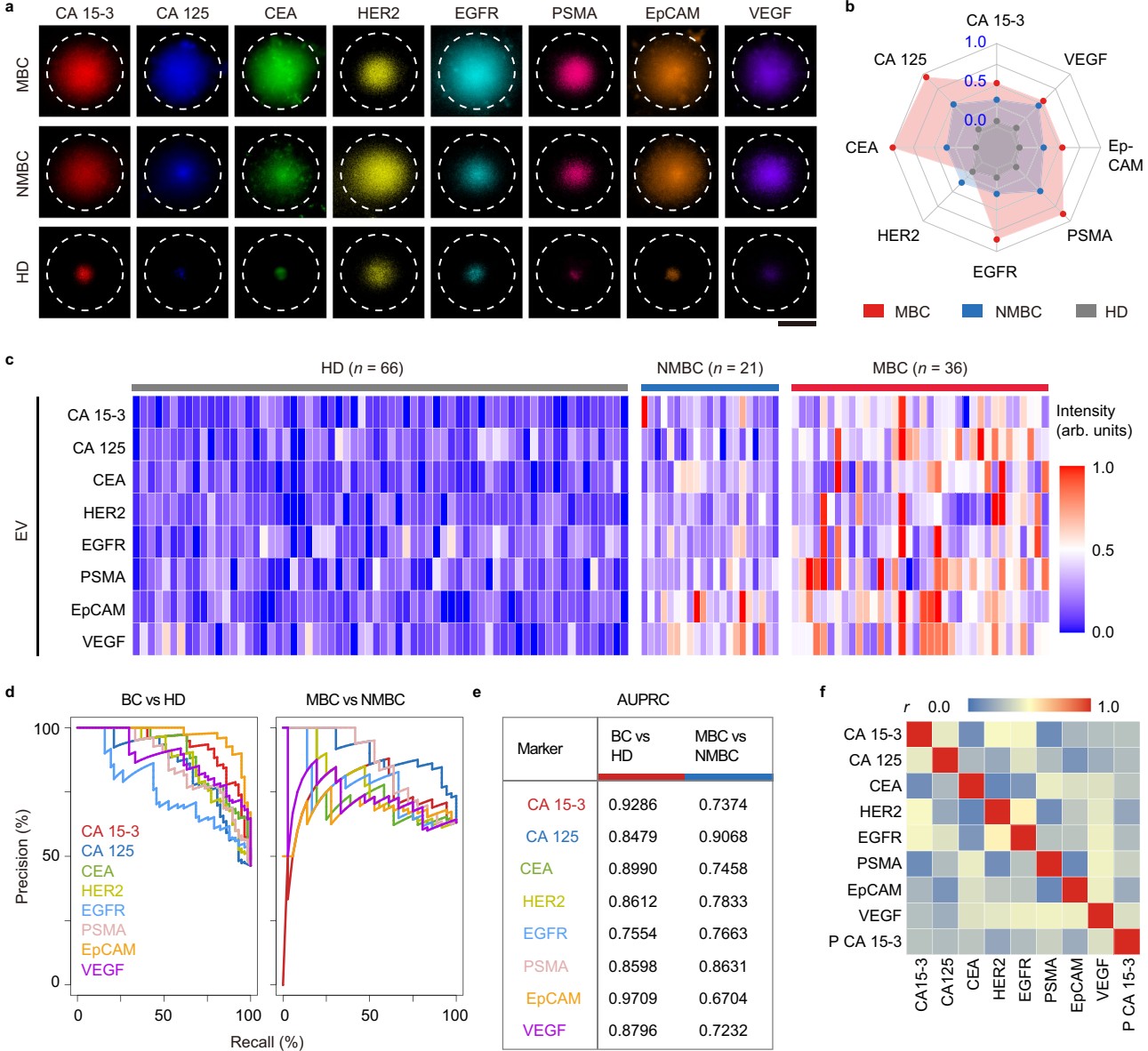

**Fig. 2 Profiling of plasma EV proteins from MBC, NMBC, and HD by TAS. a** Fluorescence images showing elevated levels of 8 EV protein markers (CA 15-3, CA 125, CEA, HER2, EGFR, PSMA, EpCAM, and VEGF) in plasma samples from a MBC patient (HR + subtype) and a NMBC patient (HER2 + subtype) compared to HD. Images are shown from a single measurement. Scale bar, 50 μm. **b** Radar plot showing TAS analyses of eight EV protein markers from MBC (red dots), NMBC (blue dots), and HD (gray dots). **c** Heatmap of EV protein profiles from 36 MBC patients, 21 NMBC patients, and 66 HD. **d** Precision-Recall Curves (PRC) for single EV protein markers to differentiate between BC and HD, as well as MBC and NMBC. **e** Summary of AUPRC (area under the PRC) of single EV protein markers for BC versus HD discrimination and MBC versus NMBC discrimination. **f** Pearson correlation matrix showing weak correlations between any pair of EV protein markers, as well as between EV markers and plasma CA 15-3. Plasma CA 15-3 is represented as P CA 15-3. Pearson correlation coefficient (*r*) is indicated. Source data are provided as a Source Data file.

PR (P124) showed a decreasing level of the $EV^M$ signature from 0 to −2, as compared to a slight increase in the concentration of plasma CA 15-3. For a HER2 + MBC patient (P112) and a metastatic TNBC patient (P45) having PD, the levels of $EV^M$ signature were elevated (Fig. 6). However, the concentration of plasma CA 15-3 remained unchanged or even decreased at the time of PD. An 80% consistency between plasma CA 15-3 and treatment response was observed when the maximum plasma CA 15-3 level was higher than twice the threshold (50 U mL$^{-1}$), whereas the consistency was decreased to 61.8% when the maximum plasma CA 15-3 level was <50 U mL$^{-1}$ (Fig. 6 and Supplementary Fig. 5). In comparison, a parallel response of $EV^M$ signature to disease status was observed in 88.6% of cases for all MBC patients. These results reveal that the $EV^M$

signature could be used for longitudinal monitoring of therapeutic responses.

**The $EV^P$ signature for predicting progression-free survival in MBC.** Furthermore, the performance of EV protein profiles in predicting clinical outcomes was investigated in a cohort of 59 MBC patients who were undergoing therapies and had baseline EV protein profiles available (Fig. 7 and Supplementary Table 13). A high level (above median value) of the $EV^P$ signature (the weighted sum of baseline intensities of 8 markers by LDA) was significantly associated with inferior progression-free survival (PFS) in Kaplan–Meier analysis (log-rank test: $P = 0.028$) (Fig. 7a). Median PFS was 475 days for the low value of $EV^P$

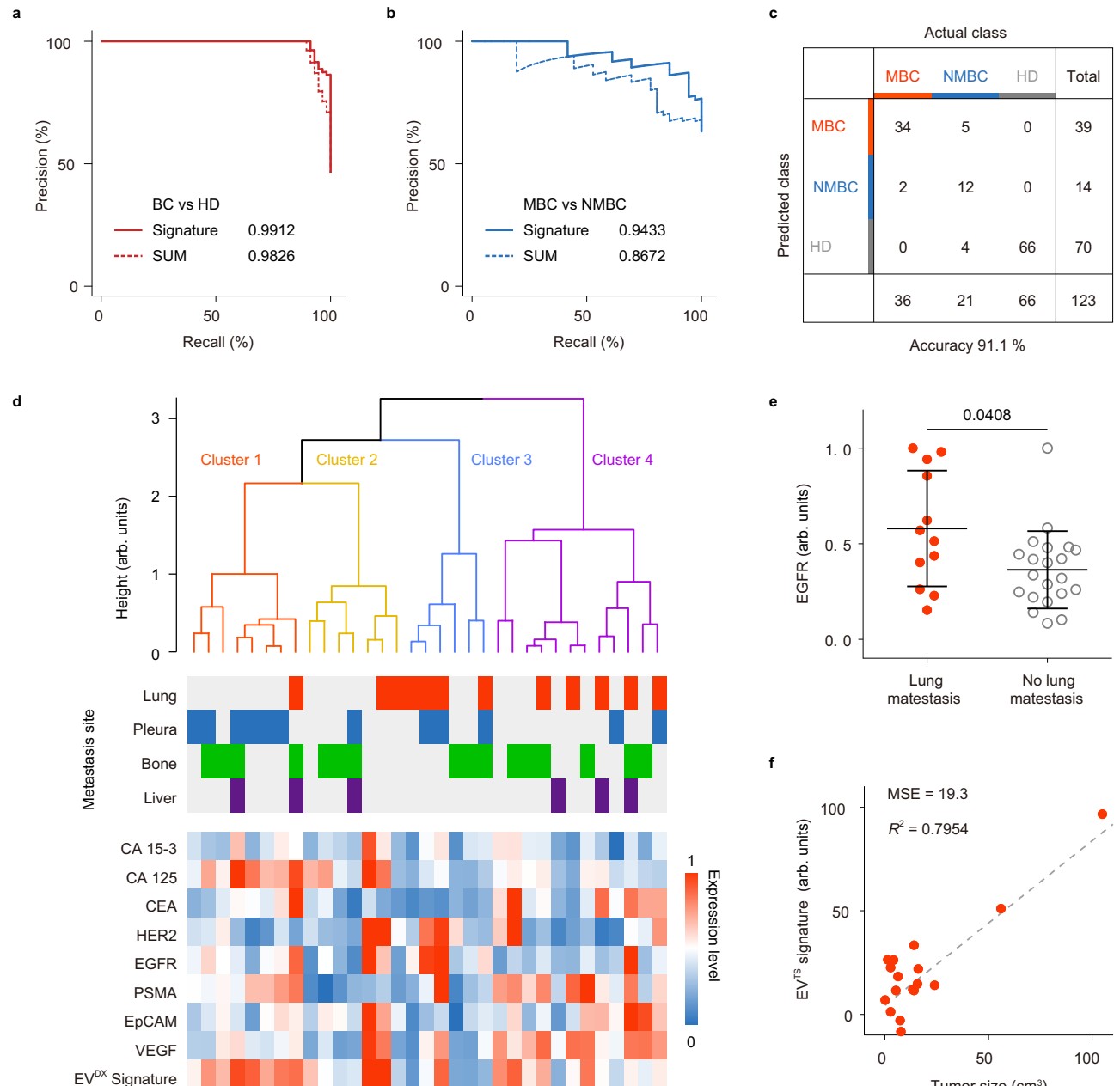

**Fig. 3 EV$^{DX}$ signature for differentiation of MBC, NMBC, and HD. a**, **b** PRC for the EV$^{DX}$ signature (a weighted sum of eight EV markers by LDA, solid lines) and SUM (unweighted sum of eight EV markers, dashed lines). **c** Confusion matrix showing an overall accuracy of 91.1% across MBC, NMBC, and HD. **d** Hierarchical clustering of individual EV protein markers and the EV$^{DX}$ signature showing no segregation according to metastasis sites of MBC patients. **e** Significant ($P = 0.0408$) elevation of EGFR level on EVs from MBC patients with lung metastasis ($n = 12$ patients, red dots), as compared to MBC patients without lung metastasis ($n = 21$ patients, gray dots). Statistical difference was determined by a two-sided, nonparametric Mann–Whitney test. $P$ value is indicated in the chart. **f** Concordance between tumor size and the EV$^{TS}$ signature identified using multivariate linear regression (MLR). Mean square errors (MSE) and R squared ($R^2$) are indicated. Linear regression result is indicated by the dashed line. Error bars represent the mean ± s.d. in (**e**). Source data are provided as a Source Data file.

signature, as compared to the median PFS of 254 days for the high value of EV$^P$ signature. Cox proportional-hazard regression analyses using a univariate model revealed that the EV$^P$ signature was a strong predictor (hazard ratio (HR) = 4.1, 95% CI = 1.1–16.4, $P = 0.0405$) of PFS in MBC. Moreover, the EV$^P$ signature remained an independent predictor (HR = 6.4, 95% CI = 1.5–27.4, $P = 0.0129$) in multivariate analysis when adjusting for age and immunohistochemical status of estrogen receptor (ER), Ki67, and HER2. In contrast, plasma CA 15-3 did not show

prognostic value in the same cohort (log-rank test: $P = 0.23$; univariate Cox regression: HR = 0.5, 95% CI = 0.2–1.6, $P = 0.239$; multivariate Cox regression: HR = 0.7, 95% CI = 0.2–2.5, $P = 0.6176$; Supplementary Fig. 6). We also noted that the best EV protein marker, PSMA alone, was associated with PFS with statistical significance (log-rank test: $P = 0.015$) and served as an independent predictor of PFS (univariate Cox regression: HR = 4.0, 95% CI = 1.2 – 13.1, $p = 0.0237$; multivariate Cox regression: HR = 4.1, 95% CI = 1.2 – 14.1, $P = 0.0277$; Fig. 7b–i). The

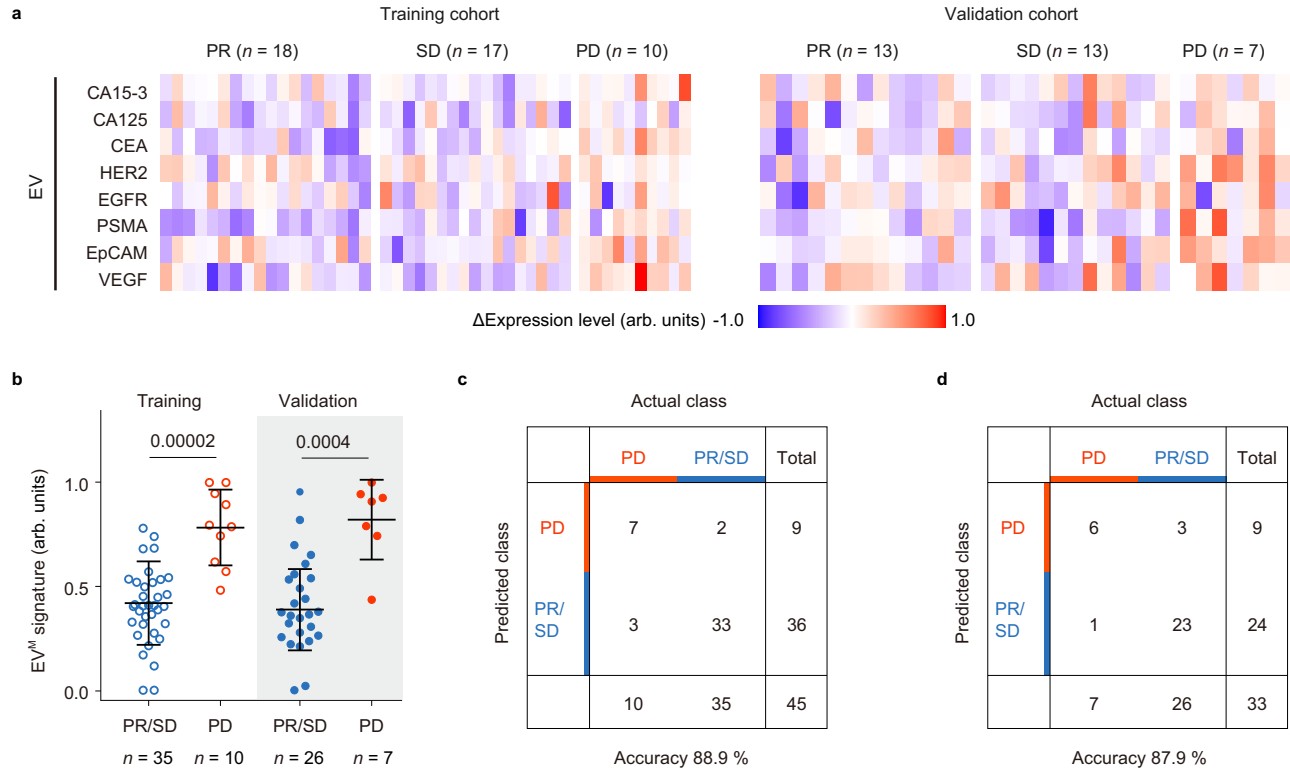

**Fig. 4 EV analyses for classification of MBC treatment response in the training and validation cohorts. a** Heatmap of ΔIntensity of EV protein markers for the training cohort involving patients with PR ($n = 18$), SD ($n = 17$), or PD ($n = 10$) and the validation cohort involving patients with PR ($n = 13$), SD ($n = 13$), or PD ($n = 7$). **b** Values of the EV$^{M}$ signature (the weighted sum of ΔIntensity of eight markers by LDA) in the training cohort (PR/SD: $n = 35$ samples, void blue dots; PD: $n = 10$ samples, void red dots) and the validation cohort (PR/SD: $n = 26$ samples, solid blue dots; PD: $n = 7$ samples, solid red dots) **c, d** Confusion matrix showing that the EV$^{M}$ signature had an accuracy of 88.9 % in differentiating PD from PR/SD for the training cohort (**c**) and 87.9% for the validation cohort (**d**). Statistical differences were determined by two-sided, nonparametric Mann–Whitney test (**b**). $P$ values are indicated in the charts. Error bars represent the mean ± s.d. in (**b**). Source data are provided as a Source Data file.

correlation between PSMA overexpression and early relapse of BC was also observed in CTCs[48]. To validate the performance of the EV$^{P}$ signature in the prognostic prediction of PFS, we further collected 16 plasma samples from MBC patients prior to treatment. This prospective cohort verified that a higher value of EV$^{P}$ signature was significantly associated with an inferior PFS in Kaplan–Meier analysis (log-rank test: $P = 0.033$ for the EV$^{P}$, Supplementary Fig. 7 and Supplementary Table 14). Collectively, these results imply that the EV$^{P}$ signature or EV PSMA may serve as an independent marker for MBC prognosis.

## Discussion

Detection of circulating EVs in the plasma of cancer patients represents a promising "liquid biopsy" and has been applied to cancer detection by using a low-cost yet sensitive and robust TAS system[31]. In this study, we demonstrate that EV protein markers measured by TAS can be used to distinguish MBC from NMBC (accuracy = 87.7%, 95% CI = 76.3–94.9%) that may otherwise be indistinguishable using other circulating biomarkers such as plasma CA 15-3, CTCs, and ctDNA[7,8,49,50]. Serial monitoring of EV protein profiles in MBC patients undergoing therapies provides a complementary approach to monitor treatment response. The EV$^{M}$ signature identified by machine learning shows an accuracy of 88.5% (95% CI = 79.2–94.6%) to discriminate PD from PR/SD across the training and validation cohorts of MBC patients. Moreover, the EV$^{P}$ signature is reported as a useful predictor of PFS for MBC patients. The point of particular interest is that EV PSMA is a significant biomarker for

monitoring and prognosis of MBC. Recent studies in exploring the value of $^{68}$Ga-PSMA-PET imaging in BC management have revealed that PSMA is overexpressed in over 90% of MBC patients[51,52]. PSMA can be constitutively internalized from the cell membrane into endosomes[53], packaged into intraluminal vesicles, and secreted to extracellular environment in EV-bound form[54,55]. Moreover, PSMA overexpression is observed in tumor-associated neovasculature, and the EVs shed from neovasculature may be prone to enter the blood circulation[54,56].

The current TAS can profile eight EV protein markers directly from 1 μL of plasma, eliminating the need for pre-isolating EVs and avoiding the interference of soluble protein. The total assay time is within 3 h and the cost is less than 1 dollar per patient (Supplementary Table 15). In comparison, detection of other liquid biopsy markers, such as CTCs or ctDNA, typically requires >7.5 mL whole blood, extensive sample preparation procedures, and sophisticated molecular analyses, making these assays impractical for clinical testing[8,57,58]. Therefore, the EV analysis we present here may have practice-changing implications for the management of cancer patients. However, the choice of protein marker candidates in this study is restricted by the availability of aptamers. To expand the marker panel, we envision that the thermophoretic sensor can be improved with the adoption of antibodies for target recognition. Moreover, the intrinsic physiological stability of antibodies allows for EV protein detection in raw plasma samples or with much less dilution folds, which may expand the utility of thermophoretic sensing to low tumor burden applications, such as residual disease monitoring and relapse prediction after neoadjuvant therapy or surgery in NMBC.

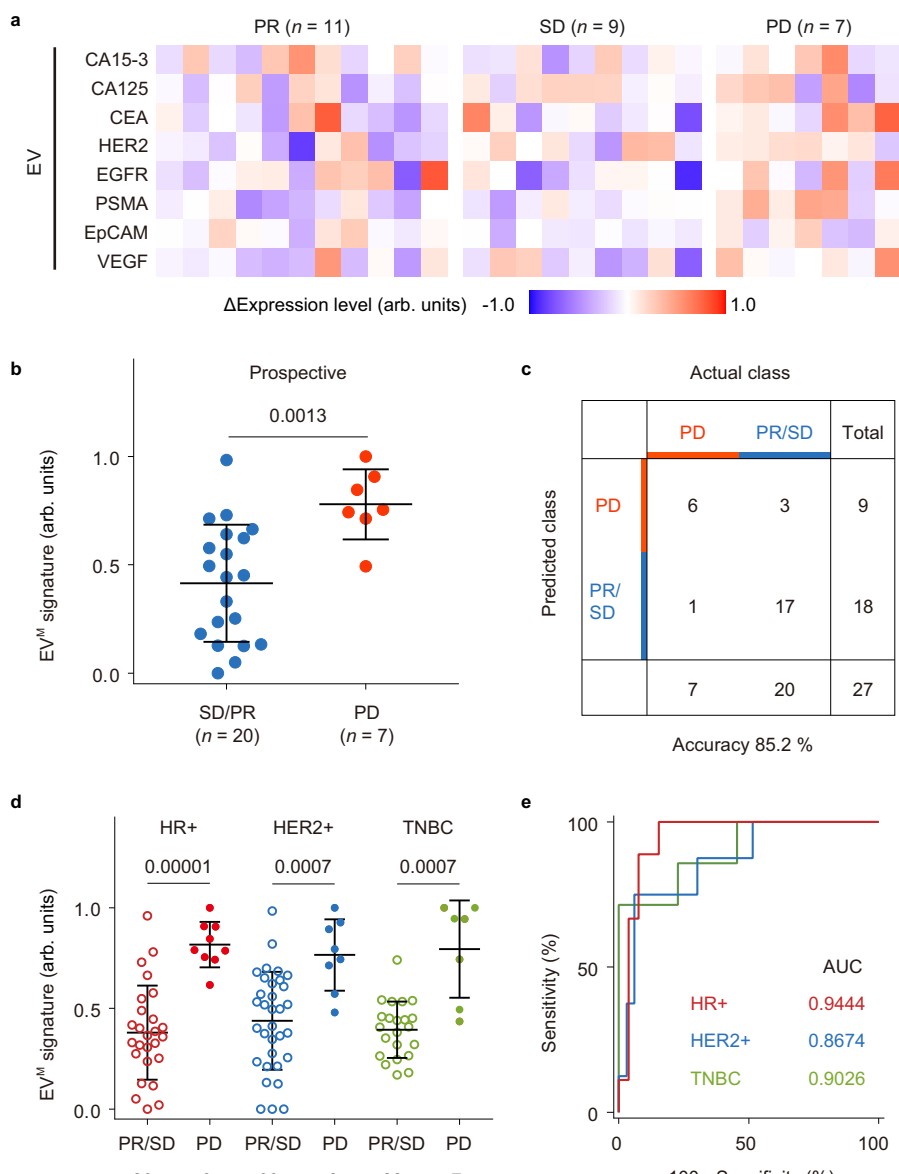

**Fig. 5 Prospective cohort of MBC patients for treatment response monitoring. a** Heatmap of ΔIntensity of EV protein markers in patients with PR ($n$ = 11), SD ($n$ = 9), or PD ($n$ = 7). **b** Values of the EV$^M$ signature (the weighted sum of ΔIntensity of eight markers by LDA) for SD/PR ($n$ = 20 samples, blue dots) and PD ($n$ = 7 samples, red dots) groups. **c** Confusion matrix showing that the EV$^M$ signature had an accuracy of 85.2% in differentiating PD from PR/SD. **d** EV$^M$ signature values in the HR + BC group (PR/SD: $n$ = 26 samples, void red dots; PD: $n$ = 9 samples, solid red dots), the HER2 + BC group (PR/SD: $n$ = 33 samples, void blue dots; PD: $n$ = 8 samples, solid blue dots), and the TNBC group (PR/SD: $n$ = 22 samples, void green dots; PD: $n$ = 7 samples, solid green dots). **e** ROC curves showing similar performance of the EV$^M$ signature for classifying treatment response in different BC subtypes (HR + , red line; HER2 + , blue line; TNBC, green line). Statistical differences were determined by two-sided, nonparametric Mann–Whitney test (**b**, **d**). P values are indicated in the charts. Error bars represent the mean ± s.d. in (**b**, **d**). Source data are provided as a Source Data file.

Finally, to achieve substantial clinical impact, studies involving rigorous validation in much larger prospective cohorts with multiyear follow-ups and the evaluation of benefit from PSMA-targeted or anti-neovasculature therapy in MBC patients stratified by EV PSMA level will be needed.

## Methods
**Study design**. The aim of this study was to investigate EV proteins as potential biomarkers for detection, monitoring, and prognostics of MBC using a thermo-phoretic aptasensor (TAS). Eight BC-associated proteins were selected as the targets in TAS measurement. As the proteins can present as both soluble and EV-bound forms in plasma samples, the capability of TAS platform in detecting EV protein markers without the interference of soluble counterparts was particularly validated. The elevation in the level of the eight EV markers was first tested using EVs derived from BC and normal mammary epithelial cell lines. In a clinical study,

286 plasma samples from MBC patients, NMBC patients, and age-matched HD were subjected to TAS measurement to obtain the profiles of eight EV markers. To establish the EV marker panel for detection, monitoring, and prognostics of MBC, the performance of individual EV markers and EV signature derived from the eight EV markers by machine learning were evaluated and compared.

**TAS procedure for clinical samples**. For surface protein profiling of plasma EVs, 1 μL clinical plasma samples were diluted by 100-folds in 1× PBS, and 10 μL diluted sample was incubated with Cy5-conjugated aptamers targeting CA 15-3, CA 125, CEA, HER2, EGFR, PSMA, EpCAM, or VEGF (0.1 μM) with MgCl$_2$ (2 mM) at 25 °C for 2 h (Supplementary Table 2). After incubation, the sample containing aptamer-labeled EVs was introduced into a microchamber with a thickness of 240 μm and a diameter of 7 mm (fabricated by sandwiching a 240-μm-thick spacer between a 1-mm-thick glass top layer and a 1-mm-thick sapphire bottom layer). The microchamber was then mounted on an inverted fluorescence microscope (DMi8, Leica, Germany) with a ×40 objective. The samples at the microchamber

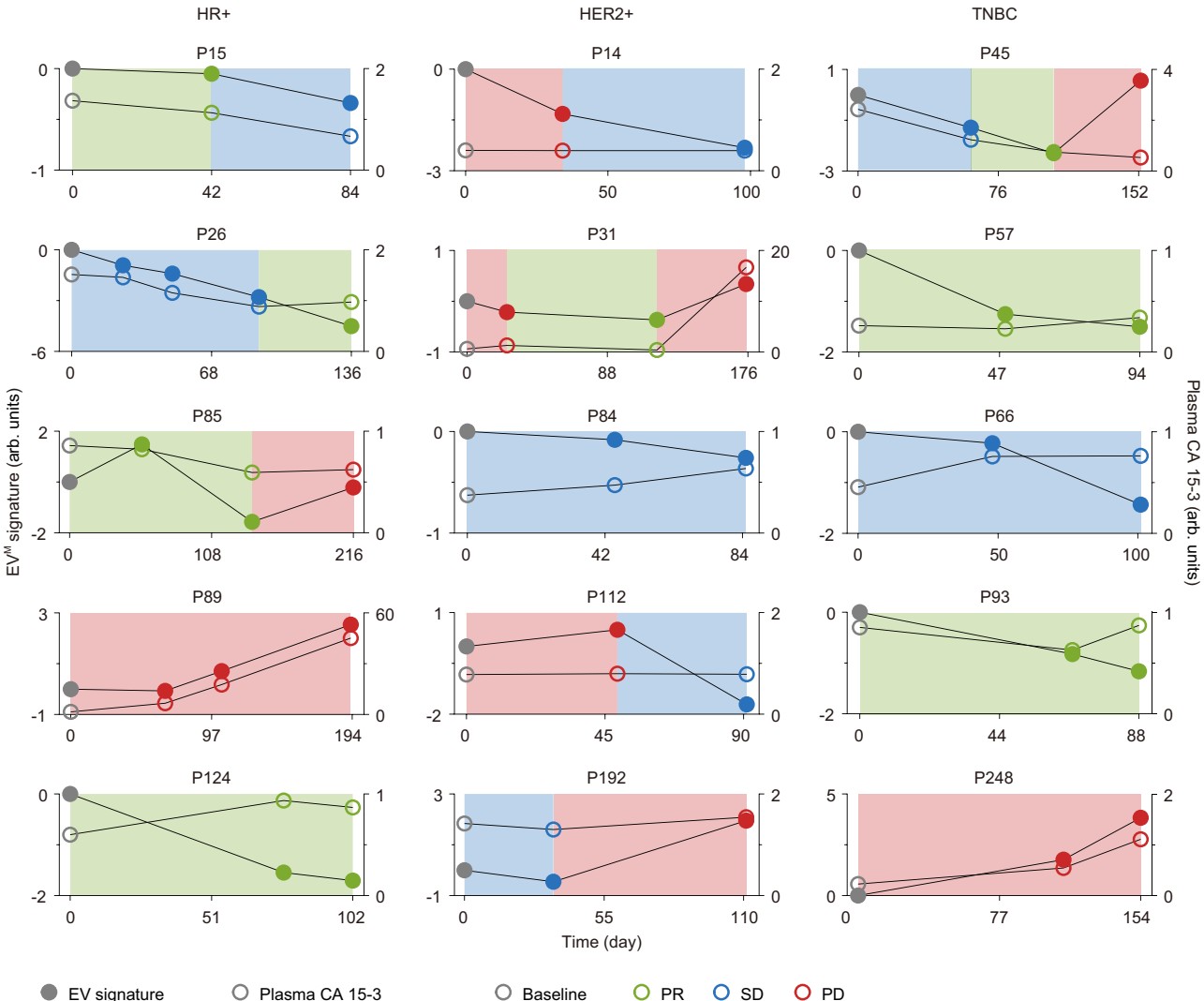

**Fig. 6 Comparison of the EV$^M$ signature and plasma CA 15-3 in longitudinal monitoring for MBC.** The EV$^M$ signature, CA 15-3 level, and therapeutic response were summarized for MBC patients in different BC subtypes ($n = 5$ for each subtype). The cumulative sum of the EV$^M$ signature at each time point was calculated and plasma CA 15-3 level was normalized by the cutoff value of 25 U mL$^{-1}$. The solid gray, green, blue, and red dots indicate the EV$^M$ signature at the baseline, PR, SD, and PD, respectively. The void gray, green, blue, and red dots indicate the plasma CA 15-3 at the baseline, PR, SD, and PD, respectively. Source data are provided as a Source Data file.

center were locally heated by a 1480-nm laser (Changchun Laser Optoelectronics Technology, China) with a power of 190 mW for 10 min to allow size-dependent enrichment of EVs. The fluorescence intensities before and after laser irradiation were captured by a sCMOS (95B, Photometrics, Canada) with $2 \times 2$ pixels binning and an exposure time of 50 ms, and recorded using ImageJ (1.52a, NIH). The experiments were performed in the dark to avoid photobleaching.

**EV titration experiments**. EVs isolated from plasma of an untreated MBC patient were suspended in 1× PBS at different concentrations ranging from $3.2 \times 10^6$ mL$^{-1}$ to $10^{10}$ mL$^{-1}$ and incubated with Cy5-conjugated single-stranded DNA (ssDNA, 0.1 μM) aptamers targeting CEA at 25 °C for 2 h. TAS was then used to detect the expression level of CEA on EVs with different concentrations. As shown in Fig. 1e, TAS exhibited a linear response for EVs in the range between $1.6 \times 10^7$ mL$^{-1}$ and $10^{10}$ mL$^{-1}$ ($R^2 = 0.93$). A limit of detection (LoD, 3× standard deviation above the blank) of $3.8 \times 10^7$ mL$^{-1}$ was obtained by TAS, which is more than $10^2$-fold lower than that of ELISA.

**TAS detection of soluble protein markers**. To determine whether soluble protein markers could be detected by TAS, three types of samples were used: (i) EV-depleted plasma diluted by 100-folds in 1× PBS; (ii) the diluted EV-depleted plasma spiked with standards of CA 15-3, CA 125, or CEA (CanAg, Sweden) at concentrations of 25 U mL$^{-1}$, 35 U mL$^{-1}$, or 5 ng mL$^{-1}$, respectively; (iii) sample (ii) spiked with EVs at $2 \times 10^9$ mL$^{-1}$ isolated from plasma of an untreated MBC patient. Both TAS and ELISA were used to detect these samples (Fig. 1d).

**Cell lines**. Human breast cancer cell lines (BT-474, SK-BR-3, and MDA-MB-231) and human mammary epithelial cell line MCF-10A were obtained from American Type Culture Collection (ATCC, USA). BT-474 cells were cultured in Dulbecco's Modified Eagle's Medium (DMEM, Gibco, USA). SK-BR-3 cells were cultured in McCoy' s5A medium (Keynentec, China). MDA-MB-231 cells were cultured in L-15 medium (Keynentec, China). MCF-10A cells were cultured in MEGM BulletKit (CC-3151 & CC-4136) medium (LONZA, USA). All media were supplemented with 10% EV-depleted fetal bovine serum (obtained by 12-h ultracentrifugation at 150,000×g) and 1% penicillin/streptomycin (Wisent, Toronto, Canada). All cell lines were incubated at 37 °C with 5% $CO_2$.

**EV isolation**. For EV isolation, the cell culture media were collected until cells reached a confluency of 70%. The collected media (300 mL) were centrifuged first at 300×g for 10 min and subsequently at 2000×g for 10 min to remove cells and large debris. The resulting supernatant was centrifuged at 10,000×g for 60 min, processed by membrane filtration (0.22 μm, Millipore, USA), and ultracentrifuged at 100,000×g for 90 min. The EVs were obtained by suspending the pellet in 1600 μL of 1× PBS. Before ultracentrifugation, the medium was packaged into eight ultracentrifuge tubes with the weight difference between every pair of tubes smaller than 0.05 g.

**EV-depleted human plasma**. In total, 1.2 mL of clinical plasma samples were diluted in 28.8 mL 1× PBS, filtered through a 0.22-μm pore filter (Millipore, USA),

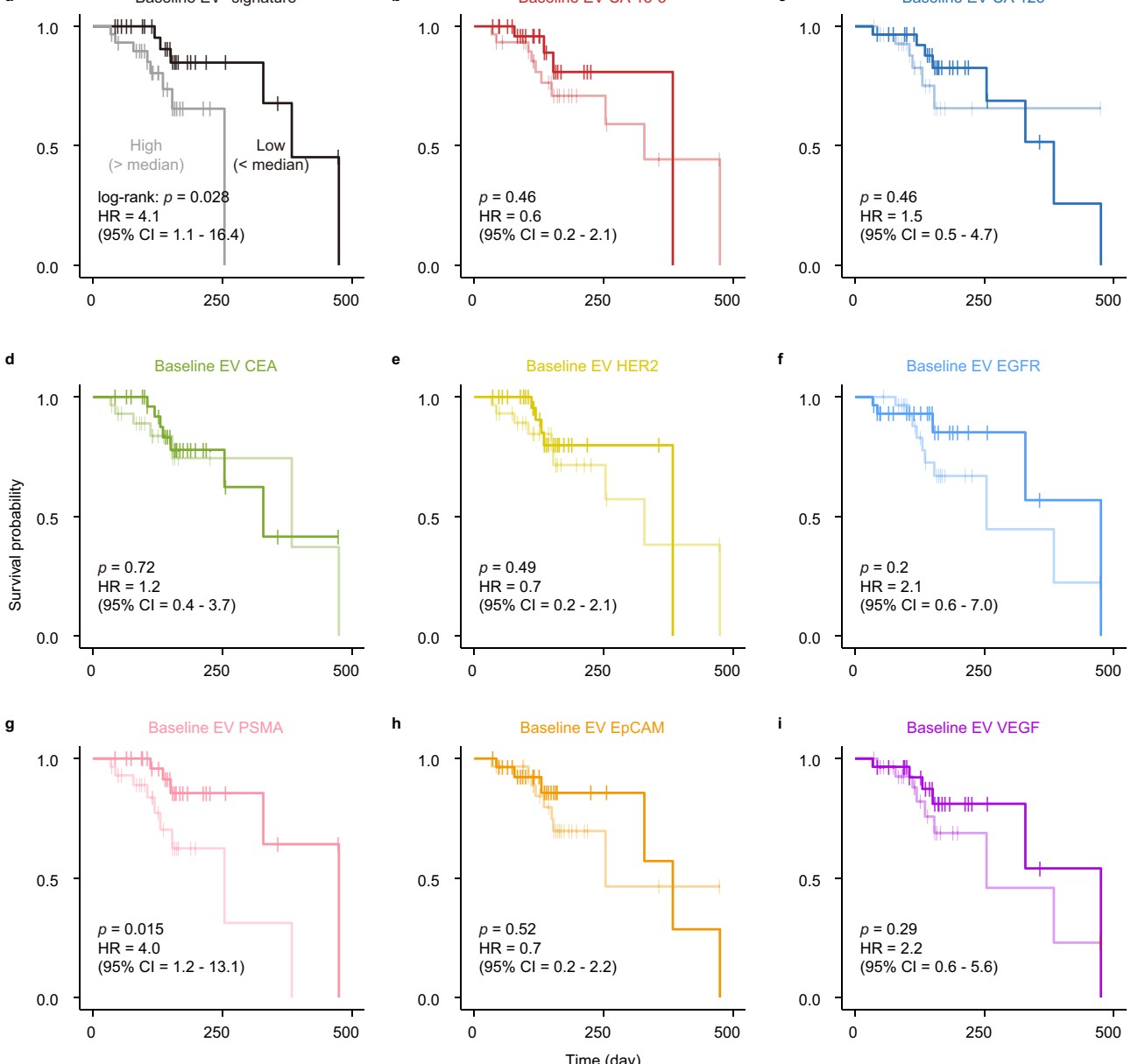

**Fig. 7 EV protein markers for prediction of progression-free survival (PFS) in a MBC cohort.** Kaplan–Meier curves showing PFS of 59 MBC patients according to the EV$^P$ signature (**a**), CA 15-3 (**b**), CA 125 (**c**), CEA (**d**), HER2 (**e**), EGFR (**f**), PSMA (**g**), EpCAM (**h**), and VEGF (**i**) on EVs before treatment (baseline). The baseline level (high or low) was stratified according to the median value. The significance of the difference was calculated by a two-sided log-rank test. Hazard ratio (HR) and 95% CI were calculated using Cox proportional-hazard regression with a univariate model. Source data are provided as a Source Data file.

then ultracentrifuged at $120,000 \times g$ at 4 °C for overnight. The EV-depleted plasma was stored at −80 °C before use.

**Nanoparticle tracking analysis (NTA)**. NTA was used to characterize the size distribution and concentration of EVs at 23 ± 2 °C (NanoSight NS300, Malvern Instrument, England). EV concentrations were adjusted to ~$10^9$ particles mL$^{-1}$ to achieve optimal counting. The data of size distribution were captured and analyzed with the NTA 3.4 Analytical Software Suite.

**Size range of EVs**. The enrichment factor of EVs by a combination of thermophoresis and convection was examined as a function of $a^2$, where $a$ is the EV diameter[59]. Therefore, large EVs were more efficiently accumulated compared to small EVs. To clarify the size range of EVs captured by our system, we first used TEM analysis and NTA analysis to characterize the size distribution of small EVs and large EVs derived from MDA-MB-231 cells. We observed that the size range and mean size of small EVs were 40–200 nm and 88 nm, and those of large EVs were 50–400 nm and 157 nm. We then applied thermophoretic enrichment to

DiO-labeled small EVs and large EVs and verified that our system can enrich both the small EVs and large EVs with the size range from 40 to 400 nm to a large extent (Supplementary Fig. 1).

**Enzyme-linked immunosorbent assay (ELISA)**. ELISA was performed to detect CA 15-3, CA 125, and CEA in EV-depleted plasma, EV-depleted plasma spiked with soluble CA 15-3, CA 125, or CEA proteins and EV-depleted plasma spiked with plasma EVs from a MBC patient, and to detect CEA in titration experiments. First, CA 15-3, CA 125, or CEA, streptavidin (SA) modified ELISA plates (96 wells, CanAg, Sweden) were coated with 100 μL mixing antibodies (125 ng mL$^{-1}$ biotin-conjugated and 50 μg mL$^{-1}$ horseradish peroxidase (HRP)-conjugated antibody) against CA 15-3 protein (CanAg, Sweden), 100 μL mixing antibodies (100 ng mL$^{-1}$ biotin-conjugated and 30 μg mL$^{-1}$ HRP-conjugated antibody) against CA 125 protein (CanAg, Sweden), or 100 μL mixing antibodies (150 ng mL$^{-1}$ biotin-conjugated and 60 μg mL$^{-1}$ HRP-conjugated antibody) against CEA protein (CanAg, Sweden). Then, 25 μL sample was added to each well. After incubation at 25 °C for 2 h (CA 15-3 and CA 125) or 1 h (CEA), each well was washed by

washing buffer for six times (CanAg, Sweden). Plates were developed with tetra-methylbenzidine (CanAg, Sweden) and stopped with stopping buffer (CanAg, Sweden). The plates were read at 405 nm with a microplate reader (Synergy H1, Biotek, USA).

ELISA was also performed to detect CD41 and CD63 on EVs isolated from plasma samples of MBC patients ($n = 5$), NMBC patients ($n = 5$), and HDs ($n = 5$) by means of ultracentrifugation. The obtained EVs were resuspended in 1× PBS at the identical volume of plasma samples. In total, 100 μL EV sample was added to CD63 or CD41 antibody modified ELISA plates (96 wells, Cusabio, China) and incubated at 37 °C for 2 h. Samples were then removed and 100 μL biotin-conjugated CD63 or CD41 antibody was added to each well. After incubation at 37 °C for 1 h, each well was washed by washing buffer three times (Cusabio, China), followed by adding 100 μL HPR-avidin and incubating at 37 °C for 1 h. Subsequently, each well was washed by washing buffer five times (Cusabio, China). Plates were developed with tetramethylbenzidine (Cusabio, China) and stopped with stopping buffer (Cusabio, China). The plates were read at 450 nm with a microplate reader (Synergy H1, Biotek, USA).

**Detection of EV CD41 and CD63 by ELISA**. We have measured the expression levels of CD41 (a platelet marker) on EVs isolated from plasma samples of MBC patients ($n = 5$), NMBC patients ($n = 5$), and HD ($n = 5$) by ELISA. However, no expression of CD41 was detected on plasma EVs from these 15 samples (Supplementary Fig. 8a). As a control, we also measured the expression of CD63 (an EV-associated marker) on plasma EVs in the same cohort by ELISA and found a higher level of EV CD63 from MBC patients than that from NMBC patients and HD (Supplementary Fig. 8b). Taken together, our assay can detect EVs in both serum and plasma samples, and the use of plasma samples can avoid interference from platelet EVs.

**Patient cohort**. Between 2019 and 2020, 220 plasma samples from BC patients were collected from the Fifth Medical Centre, Chinese PLA General Hospital, and 66 plasma samples from age-matched female HDs were collected from the Second Medical Center, Chinese PLA General Hospital. The study complied with all relevant ethical regulations and was approved by the Ethics Committee of the Fifth Medical Center of PLA General Hospital and the Chinese PLA General Hospital Ethics Committee. Informed consent to publish clinical information potentially identifying individuals was obtained from all patients. The treatment response was evaluated by computed tomography (CT) or magnetic resonance imaging (MRI). The first appearance of PD or the last censoring event for each MBC patient during the study period were recorded for the prediction of progression-free survival (PFS).

**Plasma sample collection and processing**. The clinical plasma samples were collected and processed following the Early Detection Research Network (EDRN) standard operating procedure (SOP): (i) blood samples were collected from patients into lithium-heparin plasma separator tubes (green top) and gently inverted the tube 5–8 times, (ii) the tubes were stored upright at 4 °C for 30 min, (iii) the tubes were centrifuged at 2000×$g$ for 30 min, (iv) the plasma was pipetted into labeled centrifuge tubes and stored at −80 °C before use.

**Compatibility of TAS**. To assess whether the assay compatible with both serum and plasma, we spiked the same number of MDA-MB-231 EVs ($10^{10}$ mL$^{-1}$) into EV-depleted serum and EV-depleted plasma from the same individual. We then measured the expression of eight EV markers in EV spiking samples by TAS and observed similar results between serum and plasma samples (Supplementary Fig. 9). To avoid systematic biases due to blood collection, we used the TAS assay to measure EV protein profiles in plasma samples of HD ($n = 10$) from Fifth Medical Centre, Chinese PLA General Hospital. We observed that plasma EVs from the two hospitals (Second Medical Center versus Fifth Medical Center) had similar expression levels of the 8 protein markers (Supplementary Fig. 10a). Moreover, the EV$^{DX}$ signature generated using the trained LDA model can correctly identify the HD from Fifth Medical Center (Supplementary Fig. 10b, c), demonstrating the clinical feasibility of the TAS assay.

**Repeatability and reproducibility of TAS**. Our previous studies have shown that TAS exhibited relatively low intra-batch and inter-batch variations of less than 25% for detection of serum EVs[31]. Here we assessed the repeatability and the reproducibility of TAS by measuring the expression of EV PSMA and VEGF from the same plasma sample multiple times. The TAS assay consistently achieved a relatively low intra-batch and inter-batch variations of <17% (Supplementary Fig. 11).

**Definition of EV signature**. For MBC diagnosis, the EV$^{DX}$ signature was defined as a weighted sum of the expression of eight EV markers (EV CA 15-3, CA 125, CEA, HER2, EGFR, PSMA, EpCAM, and VEGF) by a two-stage LDA. The first LDA was used to discriminate BC (including MBC and NMBC) and HD, while the second LDA was to classify the individuals predicted as BC into MBC or NMBC. The performance of each LDA was optimized and evaluated by leave-one-out

cross-validation. The EV signature against tumor size was the EV$^{TS}$ signature determined by multivariate linear regression analysis of the expression of eight EV markers. In contrast to LDA with a categorical outcome, multivariate linear regression (LR) analysis yielded a continuous outcome, thus suitable for the prediction of tumor size. For treatment response monitoring, the EV$^M$ signature was defined as a weighted sum of the relative changes in expression levels (ΔIntensity) of 8 EV markers by LDA, since the relative changes in the levels of EV markers might better reflect the treatment response (PD versus PR/SD). For prognostic prediction, the EV$^P$ signature was defined as a weighted sum of the expression of eight EV markers at baseline by LDA. LDA and LR were performed using R software (version 4.0.1).

**Assessment of the EV signature**. To assess whether any given selection of EV-associated proteins may provide similar results, we have evaluated all the combinations of 2–8 EV markers by LDA. The average accuracy for discrimination between BC and HD was increased from 85.4 to 96.8% by increasing the number of EV markers from 2 to 8. As EV-associated proteins CD63 can reflect the total number of EVs, we have further used aptamer targeting CD63 to detect plasma EVs from MBC patients ($n = 5$), NMBC patients ($n = 5$), and HD ($n = 5$) by TAS. The average expression level of CD63 on EVs from MBC patients was higher than that from NMBC patients and HD (Supplementary Fig. 12). EV CD63 achieved an overall accuracy of 73.3% for MBC diagnosis, which was lower than that of 91.1% by the EV$^{DX}$ signature. Moreover, we observed a moderate correlation between CD63 and the EV$^{DX}$ signature (Pearson correlation coefficient $r = 0.63$). These results collectively indicate that the EV signature may also reflect the number of EVs in blood plasma.

**Cross-validation**. The MBC monitoring cohort ($n = 78$) was randomly split into 100 sets of 60% training ($n = 46$) and 40% validation ($n = 32$) cohorts. As the number of PD ($n = 17$) was much smaller than that of PR/SD ($n = 61$), the sample sizes of PD and PR/SD were not balanced in the training sets. Hence, we used accuracy to evaluate the performance of classification in the training and validation sets. In the training sets, the leave-one-out cross-validation was performed to challenge the LDA model. The trained LDA model was then applied to the validation sets to verify its performance. To evaluate the degree of overfitting, we have compared the accuracy for PD versus PR/SD classification in all the 100 sets of training and validation cohorts. Both the training and validation sets showed high accuracies for classification, indicating little or no overfitting (Supplementary Fig. 13).

**EV signature weights**. As shown in Supplementary Fig. 14, the mean values of the LDA weights for eight EV markers across 100 training sets varied from −0.9 to 1.8, reflecting the variable contribution of different markers to the EV$^M$ signature. Among eight EV markers, the LDA weight was highest for EV PSMA, revealing that EV PSMA can be used to best discriminate between PD and PR/SD groups (AUC = 0.8447, 95% CI = 0.7335–0.9560). However, this value was still lower than that of the EV$^M$ signature (AUC = 0.9248, accuracy = 88.5%). In addition, we have provided the multivariate linear regression (MLR) weight of each EV marker, which varied from −57.9 for EV HER2 to 76.8 for EV CA 15-3, indicating their different contribution to the EV$^{TS}$ signature correlated with tumor size (Supplementary Fig. 15).

**Comparison of LDA and logistic regression (LR) classifiers**. The classification performance of LDA and LR in terms of MBC diagnosis, treatment response monitoring, and the prognostic prediction was compared for the same patient cohorts[60]. Similarly to LDA, the LR algorithm computed a weighted sum of the expression of 8 EV markers (the EV$^{DX}$ signature) for MBC diagnosis, a weighted sum of ΔIntensity of 8 EV markers (the EV$^M$ signature) for treatment response monitoring, and a weighted sum of the expression of 8 EV markers at baseline (the EV$^P$ signature) for prognostic prediction. Despite the limited sample size as the input, LDA and LR classifiers achieved almost the same performance for MBC diagnosis (accuracy of 91.1% versus 91.1%) and for treatment response monitoring (accuracy of 88.5% versus 89.7%) (Supplementary Table 16 and Supplementary Fig. 16). In addition, LDA had better performance than LR for prognostic prediction as indicated by a lower $P$ value (0.028 versus 0.16) and a higher hazard ratio (4.1 versus 2.2, Supplementary Fig. 16).

**Hierarchical clustering**. For the segregation of MBC patients with or without distant metastasis to a given organ, hierarchical clustering was performed to find sample groups based on similarity in EV marker expression patterns. The similarity was calculated as Pearson correlation distance between high-dimension points representing the 8 EV marker expressions and EV$^{DX}$ signature. Clustering was performed in a bottom-up manner where individual sample points served as clusters at the lowest hierarchy and closest clusters were successively merged into clusters at upper hierarchy based on the average linkage, resulting in a tree-shaped data structure. The tree was finally cut into four clusters containing MBC samples

with different patterns of metastatic site and corresponding EV marker expressions. Hierarchical clustering was performed using R software (version 4.0.1).

**TCGA assessment**. As our EV signature data for breast cancer were not available in The Cancer Genome Atlas (TCGA), we thus analyzed the expression of mRNA transcripts of coding genes of the eight protein markers across breast carcinoma tissues ($n = 1085$) and normal tissues ($n = 291$) from the TCGA TARGET GTEx cohort[61]. The mRNA expression data were analyzed by a web-based bioinformatics tool, Gene Expression Profiling Interactive Analysis, GEPIA (GEPIA 2)[62]. Among the eight markers, CA 15-3, CEA, HER2, and EpCAM showed elevated expressions at the transcriptional level in breast carcinoma tissues compared to normal tissues (Supplementary Fig. 17a). The discrepancy between the expression of EV protein markers and tissue mRNA markers highlights the need of further study for EV-based liquid biopsy.

To demonstrate the application of multivariate signature models in the TCGA TARGET GTEx database[61], we used the expression data of eight markers as inputs, and the signature generated by the LDA algorithm achieved an accuracy of 95.6% for BC versus normal classification with a PRAUC of 0.9962 (Supplementary Fig. 17). This performance was similar to that of EV$^{DX}$ signature (accuracy of 96.8% with a PRAUC of 0.9912), thus validating the combination use of eight EV protein markers in this study.

**Statistical analysis**. The intensities of individual EV protein markers from all plasma samples detected by TAS were normalized by subtracting the 2.5th percentile value and dividing by (97.5th percentile value–2.5th percentile value). SUM was calculated as the unweighted sum of the normalized intensities of eight EV markers and then normalized by the same criterion as individual EV markers. For evaluation of the performance of EV signature in MBC detection (Fig. 3a, b) and PD versus PR/SD discrimination across different cohorts or subtypes (Fig. 4b and Fig. 5b, d), EV signature was also normalized by the same criterion as individual EV markers. The upper and lower limits of the normalized intensity of individual EV markers, SUM and EV signature for MBC detection and PD versus PR/SD discrimination were 1 and 0, respectively. For convenient comparison between the consistency of EV signature and plasma CA 15-3 with treatment response in Fig. 6 and Supplementary Fig. 5, the cumulative sum of the non-normalized EV$^M$ signature was calculated for each time point. The significance of the difference between the EV surface proteins of two different sample groups in Fig. 1d was tested using a two-sided, parametric $t$ test. The significance of the difference between the EV surface protein profiles of two different patient groups was tested using a two-sided, nonparametric Mann–Whitney test. PRC was constructed to evaluate the accuracy of individual EV markers, SUM and EV$^{DX}$ signatures for detection of MBC. ROC curves were constructed to evaluate the accuracy of individual EV markers and EV$^M$ signature for therapeutic monitoring of MBC. PRC construction and AUPRC calculation were performed using R software (version 4.0.1). The $t$-SNE was performed using a perplexity parameter of 10 by MATLAB 2015b (MathWorks). Significance analyses, ROC curve construction, and AUC were performed using GraphPad Prism 7 (GraphPad). Sensitivity was defined as the probability that a test result was positive when BC, MBC, or PD was detected (true positive rate), specificity as the probability that a test result was negative for HD, NMBC, or PR/SD (true negative rate) and accuracy as the overall probability that an individual was correctly classified. The 95% CIs were calculated using a binomial distribution. The prognostic values of different variables of interest, EV signature, eight EV protein markers, and plasma CA 15-3, were analyzed using Kaplan–Meier method, log-rank test, Cox proportional-hazard regression with both univariate and multivariate (adjusted for age and immuno-histochemical status of estrogen receptor (ER), Ki67 and HER2) models. Kaplan–Meier analysis, log-rank test, and Cox proportional-hazard regression were performed using R software (version 4.0.1).

**Reporting summary**. Further information on research design is available in the Nature Research Reporting Summary linked to this article.

## Data availability

The data that support the findings of this study are available within the paper and its supplementary information files. Additional data and files are available from the corresponding author upon reasonable request. Expression data of mRNA transcripts of 8 protein markers in the TCGA TARGET GTEx cohort were accessed by publicly available database Xena (https://xenabrowser.net/datapages/?cohort=TCGA%20TARGET%20GTEx&removeHub=https%3A%2F%2Fxena.treehouse.gi.ucsc.edu%3A443). Source data are provided with this paper.

## Code availability

Code for linear discriminant analysis is available within the Supplementary Information.

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

## Acknowledgements

This work was supported financially by the National Natural Science Foundation of China (22025402, 91959101, and 21904028), Chinese Academy of Sciences (YJKYYQ20180055, YJKYYQ20190068, and ZDBS-LY-SLH025), and Beijing Talents Fund (2018000021223ZK44), and Applied Research on clinical characteristics in Beijing (Z171100001017191). Components in Fig. 1a were created with BioRender.com, and the right to re-use the clip arts was purchased.

## Author contributions

J.S. directed the research. J.S., C.L., Z.J., F.T., and S.Z. conceived the idea. F.T., S.Z., Z.H., Y.A.L., J.D., X.W., L.C., L.Q., Q.C., Y.Y., and Y.L. performed the experiments. J.S., C.L., and Y.K.L. performed the data analyses. B.D. and Y.C. assisted with data interpretation. All authors discussed the results and wrote the paper.

## Competing interests

The authors declare no competing interests.
