## [Peer Review File · Nature Communications]

Reviewers' Comments:

Reviewer #1:

Remarks to the Author:

In this paper, authors profiled cancer-associated proteins from patient plasma samples and identified eight molecular biomarkers in extracellular vesicles (EVs) for metastatic breast cancer (MBC). They developed an EV signature based on those biomarker profiles in EVs using machine learning analyses and found that the signature is able to discriminate breast cancer subtypes (MBC, non-MBC) and controls using the machine learning analyses. Also, they found that the signature can also predict the progression-free survival of MBC patients.

The manuscript is written with clear logic. However, the authors didn't provide many details on their computational analyses, especially on machine learning. Particularly, I have several major concerns as follows.

(1) Linear discrimination analysis (LDA): LDA assumes the Gaussian distribution that may not hold for the data in this study, due to limited sample size. Authors need to compare with other state-of-the-art classifiers such as logistic regression with loose assumptions.

(2) EV signature: authors didn't give details on calculating the weighted sum of the markers for getting such a signature. They first described "EV signature (the weighted sum of 8 markers using multivariate linear regression)..." on Page 13, "the EV signature (a weighted sum of 8 EV markers by LDA)" on Page 14, and then stated "The EV signature defined as the weighted sum of Δ Intensity of 8 markers by LDA" on Page 15. Please clarify.

(3) EV signature weights: Are weights sensitive to the training samples? Also, do markers contribute equally or differently to the signature based on their weights? Authors need to elaborate on the inference of the EV signature, which may provide thoughtful insights on signature-marker relations.

(4) Model evaluation: the data is very imbalanced across MBC, NMBC, and control (labels). The accuracy favors the dominant label (apparently control), and thus the AUC is biased. Authors should either balance the data or use the Precision-Recall Curves (PRC) to evaluate their classifications.

(5) t-SNE: authors need to elaborate on how to map the classification results on to a t-SNE space. First, t-SNE is unsupervised learning to cluster patients without any label information. Second, t-SNE models the local, nonlinear relationships among data samples, whereas LDA uses the linear model. Also, the t-SNE plot (Fig 3C) looks manually drawn, especially on circle-shape scattering in each cluster.

(6) Hierarchical clustering: authors used the Euclidean distance as the metric to hier. cluster patients, rather than using classical correlations. Using this metric, the clusters can be potentially biased by the markers with high expression (such as bottom markers in Fig 3D). Also, how does this clustering relate to the signature? If irrelevant, how do authors justify the capability of their EV signature for predicting metastasis to other tissues?

Minors:

(7) I suggest authors test or validate their EV signatures using public data such as TCGA.

(8) In addition to the analyses, many details on data processing are missing as well, such as expression normalization, etc.

(9) How do these markers compare to other breast cancer markers or even randomly selected proteins? Can authors provide any mechanistic insights on why these EV markers work such as signaling, functions, pathways?

Reviewer #2:

Remarks to the Author:

Fei Tian et al. previously developed an elegant approach based upon thermophoretic enrichment of aptamer-bound EVs for their quantification in 1 μ L of blood serum. In this study, the authors implement this technology to monitor breast cancer based upon the analysis of 8 pre-selected markers (CA125, CEA, HER2, EGFR, PSMA, EpCAM, VEGF) in 1 μ L of blood plasma. The specificity of the assay for EVs is demonstrated by spiking soluble proteins, which do not reveal a signal upon thermophoretic enrichment. The authors implement the assay to demonstrate that the weighted

sum of the 8 selected protein markers discriminates metastatic breast cancer from non-metastatic breast cancer and healthy donors making use of training cohorts and validation cohorts. On top the protein signature allow to monitor treatment response of metastatic breast cancer patients. The authors implement both training and validation cohorts. Overall, this study seems very promising and has the potential to advance the clinical application of EV-associated biomarkers. I however have some major comments, related to the quantification as indicated below, which should be considered by the authors.

Major comments

- The authors should provide a full explanation on the selection of the 8 protein markers. Although they refer to literature it is striking to observe that they select 8 protein markers from literature and that than the summed weight of exactly those 8 protein markers, which the authors refer to as the EV protein signature, is able to discriminate healthy donors from (non)-metastatic breast cancer and to monitor treatment response in metastatic breast cancer. This could indicate that potentially any given selection of EV-associated proteins may provide similar results, especially considering that the markers selected are not breast cancer specific. Clarifying this seems crucial to further underscore the clinical application value of this assay for breast cancer. How do results for the EV signature relate to results using aptamers targeting for example CD63, CD9 and CD81? Is the observed result a reflection in the overall changes in the total number of particles in blood plasma, which can be assessed by analyzing EV-associated proteins such as CD63, CD9 and CD81? The authors indeed indicate that EV signature correlates with tumor burden, which further indicates that Ev signature reflects the number of EVs in 1 μ L of blood plasma.

-The authors indicate in the manuscript, and refer here to the previous manuscript, that thermophoretic enrichment is size-based. This allows to specifically enrich aptamers binding EVs compared to aptamers binding soluble proteins. However, does this mean that EVs small in size (\sim 50 nm or even smaller) are less efficiently captured compared to EVs that are larger in size (\sim 100 nm or even larger)? What is the size range of EVs captured by this assay? The authors should include EM analysis (providing wide field images and close-up images) and NTA analysis to further clarify this. This becomes especially relevant when authors are comparing the 8 protein markers for EVs obtained from different cell lines, that range in size between 120 nm and 90 nm as assessed by NTA.

-Please provide further information on the results for EVs from breast cancer cell lines. How were those samples implemented in the assay? Were these samples obtained from the same number of cells? Or were same number of EVs, for example quantified by NTA, implemented for the assay ... so for example 1×10^9 particles for each cell line? The last approach seems crucial to avoid that the results reflect differences in number of EVs secreted by the different cell lines.

-In the original publication the authors studied serum EVs? Here plasma EVs? Please clarify. Is the assay compatible with both serum and plasma. Were any modifications made in the protocol to switch from serum to plasma? How do the assay results differ between serum and plasma? For example does spiking EVs from breast cancer cell lines in serum versus plasma result in similar data? How do platelet EV using CD41 marker evolve over the different samples from HD, MBC and NMBC? This is a quality control the authors can use to further support the value of the EV signature.

-The authors measure 1 μ L of blood plasma. But what is the repeatability and the reproducibility of the assay. This can be easily assessed by measuring the same sample multiple times in 1 run and in different runs over multiple time points. Maybe the authors included these experiments in the previous publication. If not, I would highly recommend to include this analysis because this is a crucial aspect in translation and assay towards a clinical application. If the authors already did this experiment, please integrate this aspect in the discussion section.

-What is the definition of breast cancer and metastatic breast cancer in this study? How do the results compare to imaging strategies? Breast cancer upon diagnosis? Metastatic breast cancer upon diagnosis? Are the cohorts of 112 and 123 samples different donors or do they partially overlap? Some clinical parameters would be valuable to include in the supplementary tables like tumor burden, tumor grade, positive lymph node status etc. In Figure 3f the EV signature shows a good correlation with tumor size? This is for the metastasis, but is this also for the primary tumor in newly diagnosed breast cancer?

Minor comments

-Define 123 samples versus 112 plasma samples. Are some of these samples similar between both groups for example healthy and non-metastatic breast cancer?

- Include a wide-field image for EM in figure 1c to provide a better idea on the heterogeneity in EV captured by the assay.
- Why is HER2 lower in MBC in figure 2a?

Reviewer #3:

Remarks to the Author:

The manuscript by Tian et al. describes a study on breast cancer detection from plasma using their recently published approach for quantitative detection of specific proteins contained in extracellular vesicles (EV). Eight nucleic acid aptamers (sequences ranging from less than 20 nucleotides to ~80 nucleotides in length) were used to detect 8 separate proteins with known expression in human breast cancer. Plasma samples from three groups of individuals were examined using this technique: 1) non-metastatic breast cancer subjects (stages I-III), 2) metastatic breast cancer subjects (stage IV), and 3) age-matched healthy donors (HD). Each of the individual markers was analyzed in each group and then a linear discriminant analysis (LDA) was used to derive an optimized weighted sum of the markers in a training and validation paradigm within the cohorts. The results of the study indicate excellent ability to discriminate healthy donors from breast cancer patients and to a lesser extent, non-metastatic from metastatic breast cancers. Further, the LDA score tracked with disease progression and outcome in the metastatic setting.

The assay itself, which generates the core data, was described in some detail in a 2019 publication in Nature BME using a partially overlapping series of aptamer probes. Laser excitation and heating produces accumulation of the aptamer-EV complex that can be readily quantified. They demonstrate many of the salient properties of the assay in the prior publication and now apply it to decent sized cohorts of the various breast cancer subjects and controls described above. The results indicate an assay that could have potential application for breast cancer detection and disease monitoring.

Issues with the manuscript:

Given the strong results and conclusions tempered by the relatively small and locally derived cohorts, the study would greatly benefit from an external validation using blinded samples. Many such cohorts are available and would make a convincing argument regarding the potential utility of this approach. This would dramatically increase the value and impact of the study since as the investigators note, there are few systemic markers for breast cancer detection or monitoring available. Without external validation, it is another in a long line of promising yet unconfirmed studies of this kind.

While it is not an absolute imperative to have complete knowledge of the binding properties of these aptamers, the casual way they are referenced as specific for the various proteins is questionable. It is at least worth describing how these aptamers were isolated with some reference to their potential specificity. Further, are these derived from published work or elsewhere?

The populations are not well defined and may be subject to systematic biases. Several points in this regard: 1) Blood from healthy donors was all collected at a different hospital/venue than the breast cancer subjects (Second Medical Center versus Fifth Medical Center). Consistent technical differences between cases and controls can have profound effects leading to false positive associations. 2) The timing of collection of the blood is unclear with respect to diagnosis, cytoreductive surgery, or other treatments. This is the case for both metastatic and non-metastatic groups. Further details should be provided, particularly related to timing of the blood draw with respect to diagnosis, treatment, and residual tumor burden.

Staging of breast cancer relates to the initial stage of the disease when diagnosed. If a cancer recurs and is metastatic, it does not become reclassified as stage IV. The description of the cohorts indicate that all metastatic cancers were stage IV which is indicative of metastatic at initial presentation. This should be verified and corrected if needed.

Jeffrey R. Marks
Duke University

Response to Reviewers

We are grateful to the reviewers for their constructive comments and insightful suggestions. Below, the reviewers' comments are shown in black, and our responses are in blue. Sections marked by red represent all altered contents in the revised manuscript. The following are our detailed responses.

Response to Reviewer #1 (machine learning, computational):

Comment: In this paper, authors profiled cancer-associated proteins from patient plasma samples and identified eight molecular biomarkers in extracellular vesicles (EVs) for metastatic breast cancer (MBC). They developed an EV signature based on those biomarker profiles in EVs using machine learning analyses and found that the signature is able to discriminate breast cancer subtypes (MBC, non-MBC) and controls using the machine learning analyses. Also, they found that the signature can also predict the progression-free survival of MBC patients.

The manuscript is written with clear logic. However, the authors didn't provide many details on their computational analyses, especially on machine learning. Particularly, I have several major concerns as follows.

Response: We thank the reviewer for the constructive review, which has helped to improve the quality of our manuscript. As suggested, we have provided more details on computational analyses. The following comments have been thoroughly addressed upon the reviewer's request.

Comment 1: Linear discrimination analysis (LDA): LDA assumes the Gaussian distribution that may not hold for the data in this study, due to limited sample size. Authors need to compare with other state-of-the-art classifiers such as logistic regression with loose assumptions.

Response: We thank the reviewer for this insightful comment. In accordance with the reviewer's request, we have compared LDA with logistic regression for the MBC diagnosis, treatment response monitoring, and prognostic prediction. We showed that LDA achieved similar classification performance as logistic regression with loose assumptions by generating the EV signature, a weighted sum of eight EV protein markers. We have included this new data in the section of "Comparison between LDA and logistic regression (LR)" in Methods, Supplementary Table 17, and Supplementary Fig. 14.

Comparison of LDA and logistic regression (LR) classifiers. The classification

performance of LDA and LR in terms of MBC diagnosis, treatment response monitoring, and prognostic prediction was compared for the same patient cohorts ⁶¹. Similarly to LDA, LR algorithm computed a weighted sum of the expression of 8 EV markers (the EV^{DX} signature) for MBC diagnosis, a weighted sum of Δ Intensity of 8 EV markers (the EV^M signature) for treatment response monitoring, and a weighted sum of the expression of 8 EV markers at baseline (the EV^P signature) for prognostic prediction. Despite the limited sample size as the input, LDA and LR classifiers achieved almost the same performance for MBC diagnosis (accuracy of 91.1 % versus 91.1 %) and for treatment response monitoring (accuracy of 88.5 % versus 89.7 %) (Supplementary Table 17 and Supplementary Fig. 14). In addition, LDA had better performance than LR for prognostic prediction as indicated by a lower *p*-value (0.028 versus 0.16) and a higher hazard ratio (4.1 versus 2.2, Supplementary Fig. 14).

Supplementary Table 17. Comparison of LDA and logistic regression (LR) classifiers in BC versus HD discrimination, MBC versus NMBC discrimination, and PR/SD versus PD discrimination (Ninety-five percent CIs are indicated in parentheses).

BC versus HD (n = 123)	LDA	LR
Sensitivity (%)	94.8 (85.4 – 98.9)	96.6 (88.1 – 99.6)
Specificity (%)	100.0 (94.6 – 100.0)	98.5 (91.8 – 100.0)
Accuracy (%)	97.6 (93.0 – 99.5)	97.6 (93.0 – 99.5)
AUC	0.9918 (0.9822 – 1.0000)	0.9957 (0.9892 – 1.0000)
MBC versus NMBC (n = 57)	LDA	LR
Sensitivity (%)	94.4 (81.3 – 99.3)	91.7 (77.5 – 98.3)
Specificity (%)	76.2 (52.8 – 91.8)	76.2 (52.8 – 91.8)
Accuracy (%)	87.7 (76.3 – 94.9)	86.0 (74.2 – 93.7)
AUC	0.9114 (0.8340 – 0.9888)	0.9140 (0.8399 – 0.9882)
PR/SD versus PD (n = 78)	LDA	LR
Sensitivity (%)	76.5 (50.1 – 93.2)	82.4 (56.6 – 96.2)
Specificity (%)	91.8 (81.9 – 97.3)	91.8 (81.9 – 97.3)
Accuracy (%)	88.5 (79.2 – 94.6)	89.7 (80.8 – 95.5)
AUC	0.9248 (0.8597 – 0.9899)	0.8944 (0.8000 – 0.9888)

Supplementary Fig. 14 | Comparison of LDA and logistic regression (LR) classifiers in MBC diagnosis, treatment response monitoring, and prognostic prediction. a, Confusion matrix showing an overall accuracy of 91.1 % across MBC, NMBC, and HD using both LDA and LR classifiers. **b,** Confusion matrix showing that LDA and LR had similar accuracies of 88.5 % and 89.7 % in differentiating PD from PR/SD. **c,** Prediction of progression-free survival (PFS) in a MBC cohort by LDA and LR. Kaplan-Meier curves showing PFS of 59 MBC patients stratified according to the median value of the EV^P signature. The significance of difference was calculated by log-rank test. Hazard ratio and 95% CI were calculated using Cox proportional-hazard regression with a univariate model.

61 Press, S. J. & Wilson, S. Choosing between Logistic Regression and Discriminant Analysis. *J. Am. Stat. Assoc.* **73**, 699-705 (1978).

Comment 2: EV signature: authors didn't give details on calculating the weighted sum of the markers for getting such a signature. They first described "EV signature (the weighted sum of 8 makers using multivariate linear regression)..." on Page 13, "the EV signature (a weighted sum of 8 EV markers by LDA)" on Page 14, and then stated "The EV signature defined as the weighted sum of Δ Intensity of 8 markers by LDA " on Page 15. Please clarify.

Response: We thank the reviewer for raising this point and we apologize for the confusion definition of the EV signature. To clarify, the EV signature for MBC diagnosis has been denoted as the EV^{DX} signature, representing a weighted sum of expression of EV CA 15-3, CA 125, CEA, HER2, EGFR, PSMA, EpCAM, and VEGF calculated by LDA. The EV signature against tumor size was the EV^{TS} signature determined by multivariate linear regression analysis of the expression of 8 EV markers. In contrast to LDA with a categorical outcome, multivariate linear regression analysis yielded a continuous outcome, thus suitable for the prediction of tumor size. The EV signature for treatment response monitoring was the EV^M signature defined as a weighted sum of Δ Intensity of 8 markers by LDA, since the relative changes in the levels of EV markers might better reflect the treatment response. The EV signature for prognostic prediction was the EV^P signature defined as a weighted sum of the expression of 8 EV markers at baseline by LDA. We have now included details regarding the definition of EV signature in the Results section and the Methods section.

Definition of EV signature. For MBC diagnosis, the EV^{DX} signature was defined as a weighted sum of the expression of 8 EV markers (EV CA 15-3, CA 125, CEA, HER2, EGFR, PSMA, EpCAM, and VEGF) by a 2-stage LDA. The first LDA was used to discriminate BC (including MBC and NMBC) and HD, while the second LDA was to classify the individuals predicted as BC into MBC or NMBC. The performance of each LDA was optimized and evaluated by leave-one-out cross-validation. The EV signature against tumor size was the EV^{TS} signature determined by multivariate linear regression analysis of the expression of 8 EV markers. In contrast to LDA with a categorical outcome, multivariate linear regression (LR) analysis yielded a continuous outcome, thus suitable for the prediction of tumor size. For treatment response monitoring, the EV^M signature was defined as a weighted sum of the relative changes in expression levels (Δ Intensity) of 8 EV markers by LDA, since the relative changes

in the levels of EV markers might better reflect the treatment response (PD versus SD/PR). For prognostic prediction, the EV^P signature was defined as a weighted sum of the expression of 8 EV markers at baseline by LDA. LDA and LR were performed using R software (version 4.0.1).

Comment 3: EV signature weights: Are weights sensitive to the training samples? Also, do markers contribute equally or differently to the signature based on their weights? Authors need to elaborate on the inference of the EV signature, which may provide thoughtful insights on signature-marker relations.

Response: We thank the reviewer for this insightful comment. To determine whether the EV signature weights are sensitive to the training samples, the MBC monitoring cohort ($n = 78$) was split into 100 balanced training (60 %) and validation (40 %) sets. As shown in Supplementary Fig. 13a, the LDA weights for 8 EV markers varied from -0.9 to 1.8, reflecting variable contribution of different markers to the EV^M signature. In addition, the LDA weights of each marker were not significantly sensitive to the training samples as indicated by the low variance in the classification performance by ROC analyses (Supplementary Fig. 13b). Among 8 EV markers, the LDA weight was highest for EV PSMA, revealing that EV PSMA can be used to best discriminate between PD and PR/SD groups (AUC = 0.8447, 95% CI = 0.7335 – 0.9560). This new data has been described in the Methods section and shown in Supplementary Fig. 13.

EV signature weights. To determine whether the EV signature weights are sensitive to the training samples, the MBC monitoring cohort ($n = 78$) was split into 100 balanced training (60 %) and validation (40 %) sets. As shown in Supplementary Fig. 13a, the LDA weights for 8 EV markers varied from -0.9 to 1.8, reflecting variable contribution of different markers to the EV^M signature. In addition, the LDA weights of each marker were not significantly sensitive to the training samples as indicated by the low variance in the classification performance by ROC analyses (Supplementary Fig. 13b). Among 8 EV markers, the LDA weight was highest for EV PSMA, revealing that EV PSMA can be used to best discriminate between PD and PR/SD groups (AUC = 0.8447, 95% CI = 0.7335 – 0.9560).

Supplementary Fig. 13 | Variable contribution of different EV markers to the EV^M signature. a, LDA weights of 8 EV markers from 100 iterations of training sets. **b**, Ensemble of ROC curves from 100 iterations of training sets showing the performance of EV^M signature in PD versus PR/SD discrimination.

Comment 4: Model evaluation: the data is very imbalanced across MBC, NMBC, and control (labels). The accuracy favors the dominant label (apparently control), and thus the AUC is biased. Authors should either balance the data or use the Precision-Recall Curves (PRC) to evaluate their classifications.

Response: We thank the reviewer for this critical comment. As requested by the reviewer, we used the Precision-Recall Curves (PRC), a better choice for imbalanced datasets, to evaluate the classification performances of the individual EV markers and the EV^{DX} signature. The new data has been described in Fig. 2d, Fig. 3a-b, and Supplementary Table 5.

Supplementary Table 5. The area under the Precision-Recall curves (AUPRC) for individual EV markers, SUM, and the EV^{DX} signature in BC versus HD discrimination as well as MBC versus NMBC discrimination.

Markers	BC versus HD (n = 123)	MBC versus NMBC (n = 57)
	AUPRC	AUPRC
CA 15-3	0.9286	0.7374
CA 125	0.8479	0.9068
CEA	0.8990	0.7458
HER2	0.8612	0.7833
EGFR	0.7554	0.7663
PSMA	0.8598	0.8631
EpCAM	0.9709	0.6704
VEGF	0.8796	0.7232
SUM	0.9825	0.8671
EV ^{DX} signature	0.9912	0.9433

Fig. 2 | d, Precision-Recall Curves (PRC) for single EV protein markers to differentiate between BC and HD, as well as MBC and NMBC. **e,** Summary of AUPRC (area under the PRC) of single EV protein markers for BC versus HD discrimination and MBC versus NMBC discrimination.

Fig. 3 | a-b, PRC for the EV^{DX} signature (a weighted sum of 8 EV markers by LDA) and SUM (unweighted sum of 8 EV markers).

Comment 5: t-SNE: authors need to elaborate on how to map the classification results on to a t-SNE space. First, t-SNE is unsupervised learning to cluster patients without any label information. Second, t-SNE models the local, nonlinear relationships among data samples, whereas LDA uses the linear model. Also, the t-SNE plot (Fig 3C) looks manually drawn, especially on circle-shape scattering in each cluster.

Response: We thank the reviewer for bringing up this point. Here the inputs of t-SNE were the two EV^{DX} signatures obtained from a two-stage LDA, in which the first EV^{DX} signature was for BC versus HD discrimination and the second EV^{DX} signature for MBC versus NMBC classification. The colors of scattering circles in the t-SNE plot indicated the patient information (MBC, NMBC, and HD) for visualization of the classification results. To avoid ambiguity, we have removed the t-SNE plot from Fig. 3.

Comment 6: Hierarchical clustering: authors used the Euclidean distance as the metric to hier. cluster patients, rather than using classical correlations. Using this metric, the clusters can be potentially biased by the markers with high expression (such as bottom markers in Fig 3D). Also, how does this clustering relate to the signature? If irrelevant, how do authors justify the capability of their EV signature for predicting metastasis to other tissues?

Response: We thank the reviewer for this insightful comment. We agree with the reviewer that the use of Euclidean distance metric for the clustering can be potentially biased by the markers with high expression. Upon the reviewer's request, we used the Pearson correlation coefficient as the metric of similarity for hierarchical clustering.

However, segregation according to metastasis sites of MBC patients was not observed based on the levels of 8 EV markers and the EV^{DX} signature (Fig. 3d). We have now clarified this point in the Results section.

Although there was heterogeneity within the 8 EV markers and the EV^{DX} signature, segregation according to metastasis sites of MBC patients was not observed by hierarchical clustering analysis (Fig. 3d). However, expression of EGFR was higher on EVs from MBC patients with lung metastasis (Fig. 3e).

Fig. 3 | d, Hierarchical clustering of individual EV protein markers and the EV^{DX} signature showing no segregation according to metastasis sites of MBC patients.

Comment 7: I suggest authors test or validate their EV signatures using public data such as TCGA.

Response: We thank the reviewer for this important comment. As our EV signature data for breast cancer were not available in The Cancer Genome Atlas (TCGA), we thus analyzed the expression of mRNA transcripts of coding genes of the 8 protein markers across breast carcinoma tissues ($n = 1085$) and normal tissues ($n = 291$) from TCGA and Genotype-Tissue Expression (GTEx) datasets. The mRNA expression data were analyzed by a web-based bioinformatics tool, Gene Expression Profiling Interactive Analysis, GEPIA. Among the 8 markers, CA 15-3, CEA, HER2, and

EpCAM showed elevated expressions at transcriptional level in breast carcinoma tissues compared to normal tissues. The discrepancy between the expression of EV protein markers and tissue mRNA markers highlights the need of further study for EV-based liquid biopsy. This new data has been described in the Methods section and shown in Supplementary Fig. 15.

TCGA assessment. As our EV signature data for breast cancer were not available in The Cancer Genome Atlas (TCGA), we thus analyzed the expression of mRNA transcripts of coding genes of the 8 protein markers across breast carcinoma tissues ($n = 1085$) and normal tissues ($n = 291$) from TCGA and Genotype-Tissue Expression (GTEx) datasets. The mRNA expression data were analyzed by a web-based bioinformatics tool, Gene Expression Profiling Interactive Analysis, GEPIA. Among the 8 markers, CA 15-3, CEA, HER2, and EpCAM showed elevated expressions at transcriptional level in breast carcinoma tissues compared to normal tissues (Supplementary Fig. 15). The discrepancy between the expression of EV protein markers and tissue mRNA markers highlights the need of further study for EV-based liquid biopsy.

Supplementary Fig. 15 | Assessment of expression levels of mRNA transcripts of 8 protein markers. The analyses were performed for breast carcinoma tissues ($n = 1085$) and normal tissues ($n = 291$) based on The Cancer Genome Atlas (TCGA) and Genotype-Tissue Expression (GTEx) datasets by Gene Expression Profiling Interactive Analysis (GEPIA). TPM, Transcripts Per Kilobase Million.

Comment 8: In addition to the analyses, many details on data processing are missing as well, such as expression normalization, etc.

Response: We thank the reviewer for pointing this out. We have now provided the details on expression normalization in the “Statistical analysis” section.

The intensities of individual EV protein markers from all plasma samples detected by TAS were normalized by subtracting the 2.5th percentile value and dividing by (97.5th percentile value – 2.5th percentile value).

Comment 9: How do these markers compare to other breast cancer markers or even randomly selected proteins? Can authors provide any mechanistic insights on why these EV markers work such as signaling, functions, pathways?

Response: We thank the reviewer for this important comment. We have measured the expression levels of CD41 (a platelet marker) on EVs isolated from plasma samples of MBC patients ($n = 5$), NMBC patients ($n = 5$), and HD ($n = 5$) by ELISA. However, no expression of CD41 was detected on plasma EVs from these 15 samples (Supplementary Fig. 8a). We also measured the expression of CD63 (an EV-associated marker) on plasma EVs in the same cohort by ELISA, and found a higher level of EV CD63 from MBC patients than that from NMBC patients and HD (Supplementary Fig. 8b). To provide mechanistic insights on the selection of CA 15-3, CA 125, CEA, HER2, EGFR, PSMA, EpCAM, and VEGF, their signaling, functions, and pathways have been provided as Supplementary Table 1.

Supplementary Fig. 8 | ELISA measurement of the expression level of EV CD41 and EV CD63 for MBC, NMBC, and HD. EVs were isolated from plasma samples of MBC patients ($n = 5$), NMBC patients ($n = 5$) and HDs ($n = 5$) by ultracentrifugation.

Supplementary Table 1. Signaling, functions, and pathways of the 8 protein markers.

Protein	Signaling pathways	Functions	Reference
CA 15-3	PI3K-AKT MEK-ERK	Survival and migration	Ref ¹
CA 125	JAK-STAT	Cell proliferation	Ref ^{2,3}
CEA	TGF- β	EMT and metastasis	Ref ⁴
HER2 & EGFR	RAS-ERK, PI3K-AKT, and mTOR	Enhance proliferation, survival, and invasion	Ref ⁵
PSMA	Integrin pathway	Neovasculature	Ref ⁶
EpCAM	Wnt	Cell cycle and enhance cell stemness	Ref ⁷
VEGF	VEGF signaling pathway	Angiogenesis and vascular permeability	Ref ⁸

1 Kufe, D. W. MUC1-C oncoprotein as a target in breast cancer: activation of signaling pathways and therapeutic approaches. *Oncogene* **32**, 1073 (2012).

2 Reinartz, S., Failer, S., Schuell, T. & Wagner, U. CA125 (MUC16) gene silencing suppresses growth properties of ovarian and breast cancer cells. *Eur. J. Cancer* **48**, 1558-1569 (2012).

3 Lakshmanan, I. *et al.* MUC16 induced rapid G2/M transition via interactions with JAK2 for increased proliferation and anti-apoptosis in breast cancer cells. *Oncogene*

31, 805-817 (2012).

4 Powell, E. *et al.* A functional genomic screen in vivo identifies CEACAM5 as a clinically relevant driver of breast cancer metastasis. *npj Breast Cancer* **4**, 9 (2018).

5 Tebbutt, N., Pedersen, M. W. & Johns, T. G. Targeting the ERBB family in cancer: couples therapy. *Nat. Rev. Cancer* **13**, 663-673 (2013).

6 Conway, R. E. *et al.* Prostate-Specific Membrane Antigen Regulates Angiogenesis by Modulating Integrin Signal Transduction. *Mol. Cell. Biol.* **26**, 5310-5324 (2006).

7 Munz, M., Baeuerle, P. A. & Gires, O. The Emerging Role of EpCAM in Cancer and Stem Cell Signaling. *Cancer Res.* **69**, 5627-5629 (2009).

8 Goel, H. L. & Mercurio, A. M. VEGF targets the tumour cell. *Nat. Rev. Cancer* **13**, 871-882 (2013).

Response to Reviewer #2 (EVs, detection):

Comment: Fei Tian et al. previously developed an elegant approach based upon thermophoretic enrichment of aptamer-bound EVs for their quantification in 1 μL of blood serum. In this study, the authors implement this technology to monitor breast cancer based upon the analysis of 8 pre-selected markers (CA125, CEA, HER2, EGFR, PSMA, EpCAM, VEGF) in 1 μL of blood plasma. The specificity of the assay for EVs is demonstrated by spiking soluble proteins, which do not reveal a signal upon thermophoretic enrichment. The authors implement the assay to demonstrate that the weighted sum of the 8 selected protein markers discriminates metastatic breast cancer from non-metastatic breast cancer and healthy donors making use of training cohorts and validation cohorts. On top the protein signature allow to monitor treatment response of metastatic breast cancer patients. The authors implement both training and validation cohorts. Overall, this study seems very promising and has the potential to advance the clinical application of EV-associated biomarkers. I however have some major comments, related to the quantification as indicated below, which should be considered by the authors.

Response: We thank the reviewer for examining our research work and providing thoughtful and insightful comments, which have helped to improve the quality of our manuscript. The following comments have been thoroughly addressed upon the reviewer's request.

Comment 1: The authors should provide a full explanation on the selection of the 8 protein markers. Although they refer to literature it is striking to observe that they select 8 protein markers from literature and that than the summed weight of exactly those 8 protein markers, which the authors refer to as the EV protein signature, is able to discriminate healthy donors from (non)-metastatic breast cancer and to monitor treatment response in metastatic breast cancer. This could indicate that potentially any given selection of EV-associated proteins may provide similar results, especially considering that the markers selected are not breast cancer specific. Clarifying this seems crucial to further underscore the clinical application value of this assay for breast cancer. How do results for the EV signature relate to results using aptamers targeting for example CD63, CD9 and CD81? Is the observed result a reflection in the overall changes in the total number of particles in blood plasma, which can be assessed by analyzing EV-associated proteins such as CD63, CD9 and CD81? The authors indeed indicate that EV signature correlates with tumor burden, which further indicates that EV signature reflects the number of EVs in 1 μL of blood plasma.

Response: We thank the reviewer for this insightful comment. In accordance with the reviewer’s request, we have provided a full explanation on the selection of the 8 protein markers in terms of signaling, functions, and pathways (Supplementary Table 1). To assess whether any given selection of EV-associated proteins may provide similar results, we have evaluated all the combinations of 2 – 8 EV markers by LDA. The average accuracy for discrimination between BC and HD was increased from 85.4 % to 96.8 %, and that for discrimination between MBC and NMBC was increased from 69.1 % to 87.7 % by increasing the number of EV markers from 2 to 8 (Supplementary Table 8). As suggested, we have further used aptamer targeting CD63 to detect plasma EVs from MBC patients ($n = 5$), NMBC patients ($n = 5$), and HD ($n = 5$) by TAS. The average expression level of CD63 on EVs from MBC patients was higher than that from NMBC patients and HD (Supplementary Fig. 12). EV CD63 achieved an overall accuracy of 73.3 % for MBC diagnosis, which was lower than that of 91.1 % by the EV^{DX} signature. Moreover, we observed a moderate correlation between CD63 and the EV^{DX} signature (Spearman correlation coefficient $r = 0.63$). These results collectively indicate that the EV signature can reflect the number of EVs in blood plasma. This new data has been described in the Methods section and shown in Supplementary Table. 8 and Supplementary Fig. 12.

Supplementary Table 1. Signaling, functions, and pathways of the 8 protein markers.

Protein	Signaling pathways	Functions	Reference
CA 15-3	PI3K-AKT MEK-ERK	Survival and migration	Ref ¹
CA 125	JAK-STAT	Cell proliferation	Ref ^{2,3}
CEA	TGF- β	EMT and metastasis	Ref ⁴
HER2 & EGFR	RAS-ERK, PI3K-AKT, and mTOR	Enhance proliferation, survival, and invasion	Ref ⁵
PSMA	Integrin pathway	Neovasculature	Ref ⁶
EpCAM	Wnt	Cell cycle and enhance cell stemness	Ref ⁷
VEGF	VEGF signaling pathway	Angiogenesis and vascular permeability	Ref ⁸

Assessment of the EV signature. To assess whether any given selection of EV-associated proteins may provide similar results, we have evaluated all the

combinations of 2 – 8 EV markers by LDA. The average accuracy for discrimination between BC and HD was increased from 85.4 % to 96.8 %, and that for discrimination between MBC and NMBC was increased from 69.1 % to 87.7 % by increasing the number of EV markers from 2 to 8 (Supplementary Table 8). As EV-associated proteins CD63 can reflect the total number of EVs, we have further used aptamer targeting CD63 to detect plasma EVs from MBC patients ($n = 5$), NMBC patients ($n = 5$), and HD ($n = 5$) by TAS. The average expression level of CD63 on EVs from MBC patients was higher than that from NMBC patients and HD (Supplementary Fig. 12). EV CD63 achieved an overall accuracy of 73.3 % for MBC diagnosis, which was lower than that of 91.1 % by the EV^{DX} signature. Moreover, we observed a moderate correlation between CD63 and the EV^{DX} signature (Spearman correlation coefficient $r = 0.63$). These results collectively indicate that the EV signature may also reflect the number of EVs in blood plasma.

Supplementary Table 8. Accuracies for BC versus HD discrimination and MBC versus NMBC discrimination by using all the combinations of 2 – 8 EV markers analyzed by LDA.

Number of EV markers	BC versus HD ($n = 123$)	MBC versus NMBC ($n = 57$)
2	85.4 %	69.1 %
3	89.0 %	72.0 %
4	90.2 %	74.6 %
5	92.3 %	77.0 %
6	93.5 %	79.1 %
7	95.4 %	83.5 %
8	96.8 %	87.7 %

Supplementary Fig. 12 | Analysis of EV CD63 by TAS. **a**, Expression levels of CD63 on plasma EVs from MBC patients ($n = 5$), NMBC patients ($n = 5$), and HD ($n = 5$). **b**, Correlation analysis between EV CD63 and the EV^{DX} signature.

Comment 2: The authors indicate in the manuscript, and refer here to the previous manuscript, that thermophoretic enrichment is size-based. This allows to specifically enrich aptamers binding EVs compared to aptamers binding soluble proteins. However, does this mean that EVs small in size (~ 50 nm or even smaller) are less efficiently captured compared to EVs that are larger in size (~100 nm or even larger)? What is the size range of EVs captured by this assay? The authors should include EM analysis (providing wide field images and close-up images) and NTA analysis to further clarify this. This becomes especially relevant when authors are comparing the 8 protein markers for EVs obtained from different cell lines, that range in size between 120 nm and 90 nm as assessed by NTA.

Response: We thank the reviewer for bringing up this important point. The enrichment factor of EVs by a combination of thermophoresis and convection was examined as a function of a^2 , where a is the EV diameter⁶⁰. Therefore, large EVs such as microvesicles were more efficiently accumulated compared to small EVs such as exosomes. To clarify the size range of EVs captured by our system, we first used TEM analysis and NTA analysis to characterize the size distribution of exosomes and microvesicles derived from MDA-MB-231 cells. We observed that the size range and mean size of exosomes were 40 – 200 nm and 88 nm, and those of microvesicles were 50 – 400 nm and 157 nm. We then applied thermophoretic enrichment to DiO-labelled exosomes and microvesicles, and verified that our system can enrich both the exosomes and microvesicles with the size range from 40 – 400 nm to a large extent (Supplementary Fig. 1). We have now included this new data in the Methods section.

Size range of EVs. The enrichment factor of EVs by a combination of thermophoresis and convection was examined as a function of a^2 , where a is the EV diameter⁶⁰. Therefore, large EVs such as microvesicles were more efficiently accumulated compared to small EVs such as exosomes. To clarify the size range of EVs captured by our system, we first used TEM analysis and NTA analysis to characterize the size distribution of exosomes and microvesicles derived from MDA-MB-231 cells. We observed that the size range and mean size of exosomes were 40 – 200 nm and 88 nm, and those of microvesicles were 50 – 400 nm and 157 nm. We then applied thermophoretic enrichment to DiO-labelled exosomes and microvesicles, and verified that our system can enrich both the exosomes and microvesicles with the size range from 40 – 400 nm to a large extent (Supplementary Fig. 1).

Supplementary Fig. 1 | Thermophoretic accumulation of exosomes and microvesicles. (a) TEM and NTA characterization of exosomes and microvesicles. Scale bars, 500 nm (insert, 250 nm). (b) Fluorescence images of DiO-labelled exosomes (10^{10} mL^{-1}) and microvesicles ($8 \times 10^9 \text{ mL}^{-1}$) after 10 min laser irradiation. Scale bar, 50 μm .

60 Baaske, P. *et al.* Extreme accumulation of nucleotides in simulated hydrothermal pore systems. *Proc. Natl. Acad. Sci. U. S. A.* **104**, 9346-9351 (2007).

Comment 3: Please provide further information on the results for EVs from breast cancer cell lines. How were those samples implemented in the assay? Were these samples obtained from the same number of cells? Or were same number of EVs, for example quantified by NTA, implemented for the assay ... so for example 1×10^9 particles for each cell line? The last approach seems crucial to avoid that the results reflect differences in number of EVs secreted by the different cell lines.

Response: We thank the reviewer for this important comment. The number of EVs from different cell lines was quantified by NTA and the same number of EVs (10^{10} mL^{-1}) was implemented in the assay. To clarify, we have included this information in the Results section and the caption of Fig. 1f.

We first characterized the performance of TAS by measuring the expression levels of the 8 protein markers on EVs derived from different BC cell lines (BT-474, SK-BR-3, and MDA-MB-231) and benign mammary epithelial cell line (MCF-10A). The number of EVs from different cell lines was quantified by NTA and the same number of EVs (10^{10} mL^{-1}) was implemented in the assay (Fig. 1f-g).

Fig. 1 | f, Fluorescence images of aptamer-labeled EVs (10^{10} mL⁻¹) after thermophoretic accumulation showing elevated levels of 8 EV protein markers (CA 15-3, CA 125, CEA, HER2, EGFR, PSMA, EpCAM, and VEGF) from the 3 BC cell lines, SK-BR-3, BT-474, and MDA-MB-231, compared to MCF-10A. Scale bar, 50 μ m.

Comment 4: In the original publication the authors studied serum EVs? Here plasma EVs? Please clarify. Is the assay compatible with both serum and plasma. Were any modifications made in the protocol to switch from serum to plasma? How do the assay results differ between serum and plasma? For example does spiking EVs from breast cancer cell lines in serum versus plasma result in similar data? How do platelet EV using CD41 marker evolve over the different samples from HD, MBC and NMBC? This is a quality control the authors can use to further support the value of the EV signature.

Response: We thank the reviewer for this critical comment. Increasing evidence has shown that the EV field is gravitating towards use of plasma compared to serum because the former is less prone to pre-analytical handling issues. Hence, we used plasma EVs in the present study without modifications made in the TAS protocol to switch from serum to plasma. To assess whether the assay compatible with both serum and plasma, we spiked the same number of MDA-MB-231 EVs (10^{10} mL⁻¹) into EV-depleted serum and EV-depleted plasma from the same individual. We then measured the expression of 8 EV markers in EV spiking samples by TAS, and observed similar results between serum and plasma samples (Supplementary Fig. 9). Upon the reviewer's request, we have further measured the expression levels of CD41 (a platelet marker) on EVs isolated from plasma samples of MBC patients ($n = 5$), NMBC patients ($n = 5$), and HD ($n = 5$) by ELISA. However, no expression of CD41 was detected on plasma EVs from these 15 samples (Supplementary Fig. 8a). As control, we also measured the expression of CD63 (an EV-associated marker) on plasma EVs in the same cohort by ELISA, and found a higher level of EV CD63 from MBC patients than that from NMBC patients and HD (Supplementary Fig. 8b). Taken together, our assay can detect EVs in both serum and plasma samples, and the use of plasma samples can avoid interference from platelet EVs. This new data has been described in the Methods section and shown in Supplementary Figs. 8-9.

Compatibility of TAS. To assess whether the assay compatible with both serum and plasma, we spiked the same number of MDA-MB-231 EVs (10^{10} mL⁻¹) into EV-depleted serum and EV-depleted plasma from the same individual. We then measured the expression of 8 EV markers in EV spiking samples by TAS, and

observed similar results between serum and plasma samples (Supplementary Fig. 9).

Enzyme-linked immunosorbent assay (ELISA). ELISA was also performed to detect CD41 and CD63 on EVs isolated from plasma samples of MBC patients ($n = 5$), NMBC patients ($n = 5$), and HDs ($n = 5$) by means of ultracentrifugation. The obtained EVs were resuspended in $1 \times$ PBS at the identical volume of plasma samples. 100 μ L EV sample was added to CD63 or CD41 antibody modified ELISA plates (96 wells, Cusabio, China) and incubated at 37 $^{\circ}$ C for 2 h. Samples were then removed and 100 μ L biotin-conjugated CD63 or CD41 antibody was added to each well. After incubation at 37 $^{\circ}$ C for 1 h, each well was washed by washing buffer for 3 times (Cusabio, China), followed by adding 100 μ L HRP-avidin and incubating at 37 $^{\circ}$ C for 1 h. Subsequently, each well was washed by washing buffer for 5 times (Cusabio, China). Plates were developed with tetramethylbenzidine (Cusabio, China) and stopped with stopping buffer (Cusabio, China). The plates were read at 450 nm with a microplate reader (Synergy H1, Biotek, USA).

Detection of EV CD41 and CD63 by ELISA. We have measured the expression levels of CD41 (a platelet marker) on EVs isolated from plasma samples of MBC patients ($n = 5$), NMBC patients ($n = 5$), and HD ($n = 5$) by ELISA. However, no expression of CD41 was detected on plasma EVs from these 15 samples (Supplementary Fig. 8a). ELISA. As control, we also measured the expression of CD63 (an EV-associated marker) on plasma EVs in the same cohort by ELISA, and found a higher level of EV CD63 from MBC patients than that from NMBC patients and HD (Supplementary Fig. 8b). Taken together, our assay can detect EVs in both serum and plasma samples, and the use of plasma samples can avoid interference from platelet EVs.

Supplementary Figure 9 | TAS measurement of EV spiking samples. MDA-MB-231 EVs (10^{10} mL⁻¹) were spiked in EV-depleted serum and EV-depleted plasma samples from the same individual, and the expression levels of 8 EV surface markers in serum and plasma were measured by TAS.

Supplementary Fig. 8 | ELISA measurement of the expression level of EV CD41 and EV CD63 for MBC, NMBC, and HD. EVs were isolated from plasma samples of MBC patients ($n = 5$), NMBC patients ($n = 5$) and HDs ($n = 5$) by ultracentrifugation.

Comment 5: The authors measure 1 μ L of blood plasma. But what is the repeatability and the reproducibility of the assay. This can be easily assessed by measuring the same sample multiple times in 1 run and in different runs over multiple time points. Maybe the authors included these experiments in the previous publication. If not, I would highly recommend to include this analysis because this is a crucial aspect in translation and assay towards a clinical application. If the authors already did this

experiment, please integrate this aspect in the discussion section.

Response: We thank the reviewer for this important comment. Our previous studies have shown that TAS exhibited relatively low intra-batch and inter-batch variations of less than 25 % for detection of serum EVs ³¹. Here we assessed the repeatability and the reproducibility of TAS by measuring the expression of EV PSMA and VEGF from the same plasma sample multiple times. The TAS assay consistently achieved a relatively low intra-batch and inter-batch variations of less than 17 % (Supplementary Fig. 11). This new data has been described in Supplementary Fig. 11.

Reproducibility of TAS. Our previous studies have shown that TAS exhibited relatively low intra-batch and inter-batch variations of less than 25 % for detection of serum EVs ³¹. Here we assessed the repeatability and the reproducibility of TAS by measuring the expression of EV PSMA and VEGF from the same plasma sample multiple times. The TAS assay consistently achieved a relatively low intra-batch and inter-batch variations of less than 17 % (Supplementary Fig. 11).

Supplementary Fig. 11 | Repeatability and Reproducibility of TAS for detecting plasma EVs. Intra-batch variation of TAS assay was determined by measuring the expression of EV PSMA and VEGF from the same plasma sample 3 times in 1 run (mean \pm s.d.). Inter-batch variation was determined by detecting plasma EVs ($n = 3$) at different time points (mean \pm s.d.). Inter- and intra-batch variations were defined as the ratio of the standard deviation to the mean value. CV, coefficient of variation.

³¹ Liu, C. *et al.* Low-cost thermophoretic profiling of extracellular-vesicle surface proteins for the early detection and classification of cancers. *Nat. Biomed. Eng.* **3**, 183-193 (2019).

Comment 6: What is the definition of breast cancer and metastatic breast cancer in this study? How do the results compare to imaging strategies? Breast cancer upon diagnosis? Metastatic breast cancer upon diagnosis? Are the cohorts of 112 and 123 samples different donors or do they partially overlap? Some clinical parameters would be valuable to include in the supplementary tables like tumor burden, tumor grade, positive lymph node status etc. In Figure 3f the EV signature shows a good correlation with tumor size? This is for the metastasis, but is this also for the primary tumor in newly diagnosed breast cancer?

Response: We thank the reviewer for this critical comment. In this study, breast cancer includes both metastatic breast cancer (MBC) and non-metastatic breast cancer (NMBC). MBC means that cancer has spread from the primary site (breast) to distant organs such as bones and lungs in the body, and NMBC means that cancer has not spread beyond the primary site. Patients received a diagnosis of MBC or NMBC based on imaging strategies and tissue biopsies were enrolled in the study. For EV-based cancer diagnosis, 123 plasma samples were collected from 36 MBC patients before salvage treatment, 21 NMBC patients before surgical therapy, and 66 healthy donors (HD). For treatment response monitoring of MBC, 112 plasma samples were collected from 53 MBC patients after 1 to 4 periods of treatments. There was no overlap between the 123 and 112 samples in these two cohorts. In accordance with the reviewer's request, more patient information about the disease (tumor burden, tumor grade, positive lymph node status, and so forth) and treatment regimens have been added to Supplementary Tables 4 and 10. As suggested, we further investigated the relationship between the EV^{TS} signature (the weighted sum of 8 makers using multivariate linear regression) and the primary tumor size in NMBC patients, and obtained a good correlation between them ($r = 0.91$, $p = 0.0001$, $n = 11$ NMBC patients, Supplementary Fig. 4). This new data has been described in the Results section and shown in Supplementary Tables 4 and 10 and Supplementary Fig. 4.

Moreover, the EV signature against tumor size (EV^{TS}, the weighted sum of 8 makers using multivariate linear regression) showed a good correlation with the sum of tumor sizes from 3 – 5 largest measurable lesions in MBC patients ($r = 0.84$, $p = 0.0001$, $n = 20$ MBC patients, Fig. 3f and Supplementary Fig. 3). In NMBC patients, the EV^{TS} signature was also correlated with the primary tumor size ($r = 0.91$, $p = 0.0001$, $n = 11$ NMBC patients, Supplementary Fig. 4).

Supplementary Table 4. Summary of MBC detection cohort.

Characteristic	MBC	NMBC	HD	Total
Total cases	36	21	66	123
Subtypes				
HR+	20	12	–	32
HER2+	8	8	–	16
TNBC	8	1	–	9
Age (year)				
Median (range)	56 (33 – 83)	55 (32 – 72)	51 (27 – 73)	52 (27 – 83)
Tumor size at first diagnosis				
T1	5	2	–	7
T2	13	10	–	23
T3	4	2	–	6
T4	3	1	–	4
Unknown	11	6	–	17
Nodal status at first diagnosis				
Node negative	8	3	–	11
N1	12	8	–	20
N2	5	0	–	5
N3	3	4	–	7
Unknown	8	6	–	14
Tumor grade at first diagnosis				
I	0	1	–	1
II	14	8	–	22
III	9	1	–	10
Unknown	13	11	–	24
Stage at first diagnosis				
I	4	2	–	6
II	16	10	–	26
III	6	6	–	12
IV	6	0	–	6
Unknown	4	3	–	7
Sampling timing (day)				
Post to diagnosis of MBC/NMBC: median (range)	6 (0 – 34)	10 (0 – 34)	–	8 (0 – 34)

Prior to neoadjuvant/salvage treatment: median (range)	2 (0 – 8)	3 (0 – 13)	–	2 (0 – 13)
Post to surgery of primary tumor: median (range)	1342 (62 – 5159)	–	–	1342 (62 – 5159)
Plasma CA 15-3 (U mL⁻¹)				
Median	42.2	9.1	7.7	9.7
Range	5.7 – 371.8	5.7 – 79.3	3.8 – 46.5	3.8 – 371.8

Supplementary Table 10. Summary of training and validation cohorts for MBC monitoring.

Characteristic	PD	SD	PR	Total
Total cases	17	30	31	78
Subtypes				
HR+	5	9	8	22
HER2+	6	14	11	31
TNBC	6	7	12	25
Age (year)				
Median (range)	48 (31 – 80)	56 (37 – 80)	55 (28 – 80)	56 (28 – 80)
Tumor size at first diagnosis				
T1	8	4	5	17
T2	2	17	14	33
T3	4	2	3	9
T4	3	4	4	11
Unknown	0	3	5	8
Nodal status at first diagnosis				
Node negative	3	8	7	18
N1	4	8	14	26
N2	7	6	3	16
N3	3	6	4	13
Unknown	0	2	3	5
Tumor grade at first diagnosis				
I	0	1	0	1
II	10	11	12	33
III	5	6	6	17
Unknown	2	12	13	27
Stage at first diagnosis				

I	2	0	1	3
II	3	15	18	36
III	8	7	6	21
IV	4	8	5	17
Unknown	0	0	1	1
Plasma CA15-3 (U mL⁻¹)				
Median	27.8	20.6	17.7	20.3
Range	6.8 – 1130	8.0 – 451	5.7 – 43.3	5.7 – 1130

Supplementary Fig. 4 | Concordance between primary tumor size and the EV^{TS} signature identified using multivariate linear regression. Spearman correlation coefficient r and p value are indicated. Linear regression result is indicated by the dashed line.

Comment 7: Define 123 samples versus 112 plasma samples. Are some of these samples similar between both groups for example healthy and non-metastatic breast cancer?

Response: We thank the reviewer for this important comment. For EV-based cancer diagnosis, 123 plasma samples were collected from 36 MBC patients before salvage treatment, 21 NMBC patients before surgical therapy, and 66 healthy donors (HD). For treatment response monitoring of MBC, 112 plasma samples were collected from MBC patients after 1 to 4 periods of treatments. To clarify, we have included this information in the Results section.

For EV-based cancer diagnosis, 123 plasma samples were collected from 36 MBC patients before salvage treatment, 21 NMBC patients before surgical therapy, and 66 age-matched HD. The expression patterns of the 8 protein markers on plasma EVs were detected by TAS (Supplementary Table 4).

Next, we assessed the ability of EV protein profiles for monitoring treatment response in a cohort of 112 plasma samples collected from MBC patients after 1 to 4 periods of treatments (Supplementary Tables 10-11).

Comment 8: Include a wide-field image for EM in figure 1c to provide a better idea on the heterogeneity in EV captured by the assay.

Response: We thank the reviewer for raising this point. We now have included a wide-field TEM image showing the heterogeneity in EVs in Fig. 1.

Fig. 1 | c, Wide field TEM image of EVs. Scale bar, 500 nm.

Comment 9: Why is HER2 lower in MBC in figure 2a?

Response: We thank the reviewer for this insightful comment. In Fig. 2a, the lower expression of EV HER2 in MBC than that in NMBC was attributed to the different molecular subtypes of breast cancer: the MBC patient was HR+ subtype and the NMBC patient was HER2+ subtype. To clarify, the subtype information has been added into the revised legend of Fig. 2a.

Fig. 2 | a, Fluorescence images showing elevated levels of 8 EV protein markers (CA 15-3, CA 125, CEA, HER2, EGFR, PSMA, EpCAM, and VEGF) in plasma samples

from a MBC patient (HR+ subtype) and a NMBC patient (HER2+ subtype) compared to HD. Scale bar, 50 μ m.

Response to Reviewer #3 (breast cancer biomarkers):

Comment: The manuscript by Tian et al. describes a study on breast cancer detection from plasma using their recently published approach for quantitative detection of specific proteins contained in extracellular vesicles (EV). Eight nucleic acid aptamers (sequences ranging from less than 20 nucleotides to ~80 nucleotides in length) were used to detect 8 separate proteins with known expression in human breast cancer. Plasma samples from three groups of individuals were examined using this technique: 1) non-metastatic breast cancer subjects (stages I-III), 2) metastatic breast cancer subjects (stage IV), and 3) age-matched healthy donors (HD). Each of the individual markers was analyzed in each group and then a linear discriminant analysis (LDA) was used to derive an optimized weighted sum of the markers in a training and validation paradigm within the cohorts. The results of the study indicate excellent ability to discriminate healthy donors from breast cancer patients and to a lesser extent, non-metastatic from metastatic breast cancers. Further, the LDA score tracked with disease progression and outcome in the metastatic setting.

The assay itself, which generates the core data, was described in some detail in a 2019 publication in Nature BME using a partially overlapping series of aptamer probes. Laser excitation and heating produces accumulation of the aptamer-EV complex that can be readily quantified. They demonstrate many of the salient properties of the assay in the prior publication and now apply it to decent sized cohorts of the various breast cancer subjects and controls described above. The results indicate an assay that could have potential application for breast cancer detection and disease monitoring.

Response: We thank the reviewer for the detailed review and the important comments, which has helped to improve the quality of our manuscript. The following comments have been thoroughly addressed upon the reviewer's request.

Comment 1: Given the strong results and conclusions tempered by the relatively small and locally derived cohorts, the study would greatly benefit from an external validation using blinded samples. Many such cohorts are available and would make a convincing argument regarding the potential utility of this approach. This would dramatically increase the value and impact of the study since as the investigators note, there are few systemic markers for breast cancer detection or monitoring available. Without external validation, it is another in a long line of promising yet unconfirmed studies of this kind.

Response: We thank the reviewer for this critical comment. We agree with the reviewer that a rigorous external validation using blinded samples will increase the

value and impact of the study. However, as the COVID-19 pandemic hinders our collaboration with other hospitals, we thereby continue to collect 27 plasma samples from MBC patients under treatment in Fifth Medical Centre, Chinese PLA General Hospital (Supplementary Table 12). In this prospective cohort, the EV^M signature was generated using the trained LDA model, which achieved an accuracy of 85.2 % (95% CI = 66.3 – 95.8 %) for discrimination between PD ($n = 7$) and PR/SD ($n = 20$) (Fig. 5 and Supplementary Table 13). We also collected 16 plasma samples from MBC patients prior to treatment to validate the performance of the EV^P signature in prognostic prediction of PFS. This new cohort verified that a higher value of EV^P signature was significantly associated with an inferior PFS in Kaplan–Meier analysis (log-rank test: $p = 0.033$, Supplementary Fig. 7). More importantly, the EV signature from the new cohorts showed similar performance compared to that of previous cohorts, suggesting the potential use of TAS assay for MBC monitoring. This new data has been described in the Results section and shown in Supplementary Tables 12-13, Fig. 5, and Supplementary Fig. 7.

In a prospective cohort of 35 plasma samples, the EV^M signature generated using the trained LDA model achieved an accuracy of 85.2 % (95% CI = 66.3 – 95.8 %) for discrimination between PD ($n = 7$) and PR/SD ($n = 20$) (Fig. 5a-c and Supplementary Tables 12-13).

To validate the performance of the EV^P signature in prognostic prediction of PFS, we further collected 16 plasma samples from MBC patients prior to treatment. This prospective cohort verified that a higher value of EV^P signature was significantly associated with an inferior PFS in Kaplan–Meier analysis (log-rank test: $p = 0.033$ for the EV^P , Supplementary Fig. 7 and Supplementary Table 15).

Fig. 5 | Prospective cohort of MBC patients for treatment response monitoring. a, Heat map of Δ Intensity of EV protein markers in patients with PR ($n = 11$), SD ($n = 9$), or PD ($n = 7$). **b,** Values of the EV signature (the weighted sum of Δ Intensity of 8 markers by LDA). **c,** Confusion matrix showing that the EV signature had an accuracy of 85.2 % in differentiating PD from PR/SD. Statistical differences were determined by two-tailed, nonparametric Mann-Whitney test (**b**). P values are indicated in the charts.

Supplementary Table 12. Summary of prospective cohort of MBC patients for treatment response monitoring.

Characteristic	PD	SD	PR	Total
Total cases	7	9	11	27
Subtypes				
HR+	4	4	5	13
HER2+	2	3	5	10
TNBC	1	2	1	4
Age (year)				
Median (range)	54 (43 – 58)	51 (43 – 63)	49 (38 – 80)	49 (38 – 80)
Tumor size at first diagnosis				
T1	0	2	1	3
T2	3	6	3	12

T3	0	0	1	1
T4	1	0	4	5
Unknown	3	1	2	6
Nodal status at first diagnosis				
Node negative	2	3	4	9
N1	1	2	4	7
N2	1	3	1	5
N3	0	0	0	0
Unknown	3	1	2	6
Tumor grade at first diagnosis				
I	0	0	0	0
II	1	3	2	6
III	2	2	3	7
Unknown	4	4	6	14
Stage at first diagnosis				
I	0	1	0	1
II	4	6	3	13
III	0	2	2	4
IV	2	0	4	6
Unknown	1	0	2	3
Plasma CA15-3 (U mL⁻¹)				
Median	42.3	37.3	16.2	24.9
Range	12 – 1482	5.9 – 1036	4.5 – 115	4.5 – 1482

Supplementary Table 13. Performance of EV markers in PR/SD versus PD discrimination for MBC patients in a prospective cohort (Ninety-five percent CIs are indicated in parentheses).

Markers	PR/SD versus PD (n = 27)			
	Sensitivity (%)	Specificity (%)	Accuracy (%)	AUC
EV CA15-3	42.9 (9.9 – 81.6)	60.0 (36.1 – 80.9)	55.6 (35.3 – 74.5)	0.5929 (0.3620 – 0.8237)
EV CA125	57.1 (18.4 – 90.1)	60.0 (36.1 – 80.9)	59.3 (38.8 – 77.6)	0.5857 (0.2676 – 0.9038)

EV CEA	71.4 (29.0 – 96.3)	70.0 (45.7 – 88.1)	70.4 (49.8 – 86.3)	0.7429 (0.5423 – 0.9434)
EV HER2	85.7 (42.1 – 99.6)	59.1 (36.4 – 79.3)	65.5 (45.7 – 82.1)	0.6857 (0.4775 – 0.8939)
EV EGFR	42.9 (9.9 – 81.6)	59.1 (36.4 – 79.3)	55.2 (51.9 – 71.3)	0.6071 (0.3594 – 0.8549)
EV PSMA	71.4 (29.0 – 96.3)	75.0 (50.9 – 91.3)	74.1 (53.7 – 88.9)	0.8143 (0.6123 – 1.0000)
EV EpCAM	57.1 (18.4 – 90.1)	70.0 (45.7 – 88.1)	66.7 (46.0 – 83.5)	0.5429 (0.2385 – 0.8472)
EV VEGF	71.4 (29.0 – 96.3)	75.0 (50.9 – 91.3)	74.1 (53.7 – 88.9)	0.8000 (0.6236 – 0.9674)
EV signature	85.7 (42.1 – 99.6)	85.0 (62.1 – 96.8)	85.2 (66.3 – 95.8)	0.8929 (0.7542 – 1.0000)

Supplementary Table 15. Summary of prospective cohort for MBC prognosis.

Characteristic	PD	Censored	Total
Total cases	5	11	16
Subtypes			
HR+	4	3	7
HER2+	0	5	5
TNBC	1	3	4
Age			
Median	43	47	46
Range	37 – 48	30 – 69	30 – 69
Baseline Plasma CA15-3 (U mL ⁻¹)			
Median	58.9	16.5	18.8
Range	4.6 – 546	5.5 – 103	4.6 – 546

Supplementary Fig. 7 | EV^P signature for prediction of progression-free survival (PFS) in a prospective cohort. Kaplan-Meier curves showing PFS of 15 MBC patients according to the EV^P signature before treatment (baseline). The baseline level (high or low) was stratified according to the median value. The significance of difference was calculated by log-rank test.

Comment 2: While it is not an absolute imperative to have complete knowledge of the binding properties of these aptamers, the casual way they are referenced as specific for the various proteins is questionable. It is at least worth describing how these aptamers were isolated with some reference to their potential specificity. Further, are these derived from published work or elsewhere?

Response: We thank the reviewer for this important comment. Aptamers with high affinity and specificity for the target are typically selected from libraries of random DNA (or RNA) sequences by systematic evolution of ligands by exponential enrichment (SELEX), which involves multiple rounds of positive selection and negative selection. In this work, aptamers binding to the extracellular regions of target proteins were selected for the detection of EV protein markers. Upon the reviewer’s request, we have included more information about aptamers such as binding affinity, targeting sites, selection method, and original published work in the Supplementary Table 3.

Supplementary Table 3. Summary of aptamer properties.

Aptamer	Affinity	Specificity	Reference
CA 15-3	$K_d = 0.135$ nM	Discrimination of variations down to single amino acid changes	Ref ⁹
CA 125	–	Binds to CA 125 and does not to CEA	Ref ¹⁰

CEA	$K_d = 5$ nM	2 order of magnitude lower affinity for BSA; discrimination between CEA and other CEACAM family members	Ref ¹¹
HER2	$K_d = 270$ nM	2 order of magnitude lower affinity for BSA	Ref ¹²
EGFR	$K_d = 56$ nM	Recognizes EGFR-positive cells (A431, U251, and U87) but does not bind to EGFR-negative cells (Jurkat).	Ref ¹³
PSMA	–	Recognize PSMA-positive C4-2 cells but not bind to PSMA-negative PC-3 cells	Ref ¹⁴
EpCAM	$K_d = 22.8$ nM	EpCAM+ Kato III; EpCAM- HEK-293T	Ref ¹⁵
VEGF	$K_d = \sim 400$ nM	–	Ref ¹⁶

9 Ferreira, C. S. M., Matthews, C. S. & Missailidis, S. DNA Aptamers That Bind to MUC1 Tumour Marker: Design and Characterization of MUC1-Binding Single-Stranded DNA Aptamers. *Tumor Biol.* **27**, 289-301 (2006).

10 Gedi, V. *et al.* Sensitive on-chip detection of cancer antigen 125 using a DNA aptamer/carbon nanotube network platform. *Sens. Actuator B-Chem.* **256**, 89-97 (2018).

11 Smith, C. L. Compositions comprising nucleic acid aptamers. U.S. patent US20130101506A1 (2006).

12 Niazi, J. H., Verma, S. K., Niazi, S. & Qureshi, A. In vitro HER2 protein-induced affinity dissociation of carbon nanotube-wrapped anti-HER2 aptamers for HER2 protein detection. *Analyst* **140**, 243-249 (2015).

13 Wang, D.-L. *et al.* Selection of DNA aptamers against epidermal growth factor receptor with high affinity and specificity. *Biochem. Biophys. Res. Commun.* **453**, 681-685 (2014).

14 Boyacioglu, O., Stuart, C. H., Kulik, G. & Gmeiner, W. H. Dimeric DNA Aptamer Complexes for High-capacity-targeted Drug Delivery Using pH-sensitive Covalent Linkages. *Mol. Ther.-Nucl. Acids* **2** (2013).

15 Song, Y. *et al.* Selection of DNA Aptamers against Epithelial Cell Adhesion Molecule for Cancer Cell Imaging and Circulating Tumor Cell Capture. *Anal. Chem.* **85**, 4141-4149 (2013).

16 Potty, A. S. R. *et al.* Biophysical characterization of DNA aptamer interactions with vascular endothelial growth factor. *Biopolymers* **91**, 145-156 (2009).

Comment 3: The populations are not well defined and may be subject to systematic biases. Several points in this regard: 1) Blood from healthy donors was all collected at a different hospital/venue than the breast cancer subjects (Second Medical Center

versus Fifth Medical Center). Consistent technical differences between cases and controls can have profound effects leading to false positive associations. 2) The timing of collection of the blood is unclear with respect to diagnosis, cytoreductive surgery, or other treatments. This is the case for both metastatic and non-metastatic groups. Further details should be provided, particularly related to timing of the blood draw with respect to diagnosis, treatment, and residual tumor burden.

Response: We thank the reviewer for bringing up these important points. To address the reviewer's concern, we used the TAS assay to measure EV protein profiles in plasma samples of HD ($n = 10$) from Fifth Medical Centre, Chinese PLA General Hospital. We observed that plasma EVs from the two hospitals (Second Medical Center versus Fifth Medical Center) had the similar expression levels of the 8 protein markers (Supplementary Fig. 10a). Moreover, the EV^{DX} signature generated using the trained LDA model can correctly identify the HD from Fifth Medical Center (Supplementary Fig. 10b-c), demonstrating the clinical feasibility of TAS assay. In accordance with the reviewer's request, we have provided the timing of collection of the blood with respect to diagnosis, treatment, and residual tumor burden for both MBC and NMBC groups (Supplementary Table 4).

To avoid systematic biases due to blood collection, we used the TAS assay to measure EV protein profiles in plasma samples of HD ($n = 10$) from Fifth Medical Centre, Chinese PLA General Hospital. We observed that plasma EVs from the two hospitals (Second Medical Center versus Fifth Medical Center) had the similar expression levels of the 8 protein markers (Supplementary Fig. 10a). Moreover, the EV^{DX} signature generated using the trained LDA model can correctly identify the HD from Fifth Medical Center (Supplementary Fig. 10b-c), demonstrating the clinical feasibility of TAS assay.

Supplementary Fig. 10 | TAS assay showing similar expression levels of EV surface proteins for plasma samples from two different hospitals. a, Expression levels of 8 EV surface proteins in plasma samples collected from Fifth Medical Center ($n = 10$, mean \pm s.d.) and Second Medical Center ($n = 66$, mean \pm s.d.). Statistical differences were determined by Mann-Whitney test. **b-c**, Values of EV^{DX} signature (**b**) and t-SNE plot (**c**) showing that all the HDs from Fifth Medical Center can be correctly differentiated from BC patients.

Supplementary Table 4. Summary of MBC detection cohort.

Characteristic	MBC	NMBC	HD	Total
Total cases	36	21	66	123
Subtypes				
HR+	20	12	–	32
HER2+	8	8	–	16
TNBC	8	1	–	9
Age (year)				
Median (range)	56 (33 – 83)	55 (32 – 72)	51 (27 – 73)	52 (27 – 83)
Tumor size at first diagnosis				
T1	5	2	–	7
T2	13	10	–	23
T3	4	2	–	6

T4	3	1	–	4
Unknown	11	6	–	17
Nodal status at first diagnosis				
Node negative	8	3	–	11
N1	12	8	–	20
N2	5	0	–	5
N3	3	4	–	7
Unknown	8	6	–	14
Tumor grade at first diagnosis				
I	0	1	–	1
II	14	8	–	22
III	9	1	–	10
Unknown	13	11	–	24
Stage at first diagnosis				
I	4	2	–	6
II	16	10	–	26
III	6	6	–	12
IV	6	0	–	6
Unknown	4	3	–	7
Sampling timing (day)				
Post to diagnosis of MBC/NMBC: median (range)	6 (0 – 34)	10 (0 – 34)	–	8 (0 – 34)
Prior to neoadjuvant/salvage treatment: median (range)	2 (0 – 8)	3 (0 – 13)	–	2 (0 – 13)
Post to surgery of primary tumor: median (range)	1342 (62 – 5159)	–	–	1342 (62 – 5159)
Plasma CA 15-3 (U mL⁻¹)				
Median	42.2	9.1	7.7	9.7
Range	5.7 – 371.8	5.7 – 79.3	3.8 – 46.5	3.8 – 371.8

Comment 4: Staging of breast cancer relates to the initial stage of the disease when diagnosed. If a cancer recurs and is metastatic, it does not become reclassified as stage IV. The description of the cohorts indicate that all metastatic cancers were stage IV which is indicative of metastatic at initial presentation. This should be verified and corrected if needed.

Response: We thank the reviewer for this important comment. We totally agree with the reviewer that staging of breast cancer relates to the initial stage of the disease when diagnosed. Here MBC included both stage IV breast cancer and metastatic breast cancer with the initial stage of I-III. To avoid the ambiguity, we have corrected the information about staging of breast cancer throughout the entire manuscript. Please also see the response to Comment 6, Reviewer #2 for the staging information.

Supplementary Data. Additional patient information. ▲ indicates the time point for plasma sampling.

Patient	Status	Gender	Age	Subtype	Baseline	Treatment-1	Treatment-2	Treatment-3	Treatment-4
2	MBC	Female	33	HR+ HER2-	Post-surgery, 8 weeks MBC recurrence, 0 day Prior to salvage therapy ▲				
15	MBC	Female	37	HR+ HER2-	MBC diagnosis, 4 weeks Prior to salvage therapy ▲	paclitaxel + capecitabine 6 weeks, PR ▲	paclitaxel + capecitabine 6 weeks, SD ▲		
26	MBC	Female	63	HR+ HER2-	Post-surgery, 362 weeks MBC recurrence, 4 weeks Prior to salvage therapy ▲	paclitaxel + capecitabine 4 weeks, SD ▲	paclitaxel + capecitabine 4 weeks, SD ▲	paclitaxel + capecitabine 6 weeks, SD ▲	paclitaxel + capecitabine 6 weeks, PR ▲
29	MBC	Female	50	HR+ HER2-	MBC recurrence, 0 day Prior to salvage therapy ▲	paclitaxel 14 weeks, SD ▲			
35	MBC	Female	48	HR+ HER2-	MBC recurrence, 0 day Prior to salvage therapy ▲	letrozole + leuprorelin 32 weeks, SD ▲			
38	MBC	Female	70	HR+ HER2-	Prior to salvage therapy ▲	fulvestrant 7 weeks, PR ▲			
39	MBC	Female	72	HR+ HER2-	Prior to salvage therapy ▲	fulvestrant 11 weeks, PR ▲			
49	MBC	Female	49	HR+ HER2-	Post-surgery, 527 weeks MBC recurrence, 0 day Prior to salvage therapy ▲				
50	MBC	Female	60	HR+ HER2-	Post-surgery, 737 weeks MBC recurrence, 12 days Prior to salvage therapy ▲	fulvestrant 9 weeks, PR ▲			
51	MBC	Female	26	HR+ HER2-	Prior to salvage therapy Prior to salvage therapy ▲	letrozole + palbociclib 9 weeks, SD ▲	letrozole + palbociclib 14 weeks, SD		
56	NMBC	Female	61	HR+ HER2-	NMBC diagnosis, 15 days Prior to neoadjuvant therapy ▲				
60	MBC	Female	46	HR+ HER2-	MBC diagnosis, 28 days	paclitaxel + capecitabine	paclitaxel + capecitabine		

					Prior to salvage therapy ▲	7 weeks, PR ▲	6 weeks, PR ▲		
63	MBC	Female	56	HR+ HER2-	Prior to salvage therapy ▲	apatinib 11 weeks, PD ▲			
74	NMBC	Female	64	HR+ HER2-	NMBC diagnosis, 34 days Prior to neoadjuvant therapy ▲				
76	MBC	Female	45	HR+ HER2-	MBC diagnosis, 5 days Prior to salvage therapy ▲	paclitaxel + zoledronic acid 5 weeks, PR ▲			
85	MBC	Female	48	HR+ HER2-	Post-surgery, 458 weeks MBC recurrence, 7 days Prior to salvage therapy ▲	paclitaxel 8 weeks, PR ▲	paclitaxel + capecitabine 12 weeks, PR ▲	capecitabine 11 weeks, PD ▲	
86	MBC	Female	38	HR+ HER2-	Post-surgery, 77 weeks MBC recurrence, 1 day Prior to salvage therapy ▲	paclitaxel + doxorubicin 7 weeks, SD ▲	paclitaxel + doxorubicin 15 weeks, PR ▲		
89	MBC	Female	33	HR+ HER2-	Prior to salvage therapy ▲	vinorelbine + pyrotinib 9 weeks, PD ▲	paclitaxel 6 weeks, PD ▲	apatinib 13 weeks, PD ▲	
90	MBC	Female	43	HR+ HER2-	Prior to salvage therapy ▲	goserelin + letrozole + palbociclib 7 weeks, SD ▲	goserelin + letrozole + palbociclib 5 weeks, PD ▲	bevacizumab + vinorelbine 5 weeks, SD ▲	
92	NMBC	Female	51	HR+ HER2-	NMBC diagnosis, 6 days Prior to neoadjuvant therapy ▲				
96	MBC	Female	65	HR+ HER2-	Post-surgery, 413 weeks MBC recurrence, 0 day Prior to salvage therapy ▲				
97	MBC	Female	38	HR+ HER2-	Prior to salvage therapy ▲	paclitaxel 6 weeks, PD ▲			
100	MBC	Female	61	HR+ HER2-	Post-surgery, 549 weeks MBC recurrence, 0 day Prior to salvage therapy ▲	paclitaxel 7 weeks, SD ▲	paclitaxel 24 week, SD		
102	MBC	Female	56	HR+ HER2-	Prior to salvage therapy ▲	fulvestrant 26 weeks, SD			

111	MBC	Female	56	HR+ HER2-	MBC diagnosis, 22 days Prior to salvage therapy ▲	paclitaxel + capecitabine 7 weeks, SD ▲	paclitaxel + capecitabine 7 weeks, SD		
114	MBC	Female	47	HR+ HER2-	Prior to salvage therapy ▲	paclitaxel 7 weeks, SD ▲	anastrozole + goserelin 68 weeks, PR		
116	MBC	Female	43	HR+ HER2-	Prior to salvage therapy ▲	capecitabine 25 weeks, PR ▲	capecitabine 7 weeks, PD ▲		
122	MBC	Female	46	HR+ HER2-	Post-surgery, 100 weeks MBC recurrence, 22 days Prior to salvage therapy ▲	anastrozole + leuprorelin 6 weeks, SD ▲	anastrozole + leuprorelin 22 weeks, SD		
127	MBC	Female	65	HR+ HER2-	Post-surgery, 563 weeks MBC recurrence, 26 days Prior to salvage therapy ▲				
132	MBC	Female	47	HR+ HER2-	Prior to salvage therapy ▲	paclitaxel + capecitabine 23 weeks, SD			
133	NMBC	Female	32	HR+ HER2-	NMBC diagnosis, 24 days Prior to neoadjuvant therapy ▲				
155	MBC	Female	52	HR+ HER2-	Post-surgery, 159 weeks MBC recurrence, 13 days Prior to salvage therapy ▲	anastrozole + goserelin 55 weeks, PD			
159	MBC	Female	38	HR+ HER2-	Post-surgery, 154 weeks MBC recurrence, 17 days Prior to salvage therapy ▲				
164	MBC	Female	64	HR+ HER2-	Post-surgery, 531 weeks MBC recurrence, 2 days Prior to salvage therapy ▲				
165	MBC	Female	68	HR+ HER2-	Prior to salvage therapy ▲	paclitaxel 14 weeks, SD			
168	NMBC	Female	62	HR+ HER2-	NMBC diagnosis, 12 days Prior to neoadjuvant therapy ▲				
210	MBC	Female	64	HR+ HER2-	Prior to salvage therapy ▲	paclitaxel			

						17 weeks, SD			
211	MBC	Female	47	HR+ HER2-	Post-surgery, 279 weeks MBC recurrence, 3 days Prior to salvage therapy ▲	paclitaxel + capecitabine 9 weeks, SD	vinorelbine 15 weeks, SD		
271	NMBC	Female	55	HR+ HER2-	NMBC diagnosis, 15 days Prior to neoadjuvant therapy ▲				
288	MBC	Female	48	HR+ HER2-	Prior to salvage therapy ▲	paclitaxel 13 weeks, SD	toremifene 28 weeks, PD		
344	MBC	Female	54	HR+ HER2-	Prior to salvage therapy ▲	paclitaxel 6 weeks, PD ▲			
345	NMBC	Female	40	HR+ HER2-	NMBC diagnosis, 2 days Prior to neoadjuvant therapy ▲				
357	NMBC	Female	46	HR+ HER2-	NMBC diagnosis, 0 day Prior to neoadjuvant therapy ▲				
414	NMBC	Female	52	HR+ HER2-	NMBC diagnosis, 7 days Prior to neoadjuvant therapy ▲				
415	NMBC	Female	32	HR+ HER2-	NMBC diagnosis, 14 days Prior to neoadjuvant therapy ▲				
449	MBC	Female	48	HR+ HER2-	Prior to salvage therapy ▲	anastrozole + goserelin 7 weeks, PD			
503	MBC	Female	50	HR+ HER2-	Post-surgery, 86 weeks MBC recurrence, 21 days Prior to salvage therapy ▲	paclitaxel 25 weeks, SD			
517	MBC	Female	37	HR+ HER2-	Prior to salvage therapy ▲	fulvestrant + goserelin 12 weeks, PD			
531	NMBC	Female	49	HR+ HER2-	NMBC diagnosis, 0 day Prior to neoadjuvant therapy ▲	letrozole 7 weeks, SD			
581	MBC	Female	57	HR+ HER2-	Post-surgery, 615 weeks MBC recurrence, 14 days Prior to salvage therapy ▲				

621	MBC	Female	43	HR+ HER2-	Prior to salvage therapy ▲	paclitaxel 8 weeks, SD	vinorelbine 26 weeks, PD		
681	MBC	Female	41	HR+ HER2-	MBC diagnosis, 18 days Prior to salvage therapy ▲	paclitaxel 31 weeks, PR	anastrozole + leuprorelin 6 weeks, SD		
704	NMBC	Female	61	HR+ HER2-	NMBC diagnosis, 2 days Prior to neoadjuvant therapy ▲				
18	MBC	Female	56	HR+ HER2+	Prior to salvage therapy ▲	capecitabine + trastuzumab 12 weeks, SD ▲	capecitabine + trastuzumab 26 weeks, PR		
31	MBC	Female	59	HR+ HER2+	Prior to salvage therapy ▲	anastrozole + trastuzumab 4 weeks, PD ▲	capecitabine + pyrotinib 13 weeks, PR ▲	capecitabine + pyrotinib 8 weeks, PD ▲	
40	MBC	Female	48	HR+ HER2+	Prior to salvage therapy ▲	anastrozole + goserelin + lapatinib 16 weeks, SD			
46	MBC	Female	56	HR+ HER2+	Post-surgery, 192 weeks MBC recurrence, 4 days Prior to salvage therapy ▲	fulvestrant + lapatinib 9 weeks, SD ▲	fulvestrant + lapatinib 7 weeks, SD ▲		
67	MBC	Female	32	HR+ HER2+	Prior to salvage therapy ▲	capecitabine + lapatinib 13 weeks, SD			
94	MBC	Female	56	HR+ HER2+	Prior to salvage therapy ▲	capecitabine + lapatinib 6 weeks, SD ▲	capecitabine + lapatinib 18 weeks, SD		
124	MBC	Female	59	HR+ HER2-	Prior to salvage therapy ▲	paclitaxel + cyclophosphamide 11 weeks, PR ▲	paclitaxel + cyclophosphamide 4 weeks, PR ▲		
142	MBC	Female	78	HR+ HER2+	Prior to salvage therapy ▲	trastuzumab + lapatinib 4 weeks, SD			
154	MBC	Female	56	HR+ HER2+	Post-surgery, 135 weeks MBC recurrence, 12 days Prior to salvage therapy ▲				
201	MBC	Female	54	HR+ HER2+	Prior to salvage therapy ▲	trastuzumab 44 weeks, PR			

212	MBC	Female	19	HR+ HER2+	Prior to salvage therapy ▲	anastrozole + goserelin + trastuzumab + platinum 7 weeks, SD ▲			
246	MBC	Female	43	HR+ HER2+	Prior to salvage therapy ▲	pyrotinib 18 weeks, PD			
258	MBC	Female	52	HR+ HER2+	Prior to salvage therapy ▲	paclitaxel + trastuzumab 17 weeks, PR ▲	paclitaxel + trastuzumab 4 weeks, SD ▲		
272	MBC	Female	44	HR+ HER2+	Prior to salvage therapy ▲	paclitaxel + lapatinib 14 weeks, SD ▲	paclitaxel + lapatinib 33 weeks, PD		
273	MBC	Female	50	HR+ HER2+	Prior to salvage therapy ▲	paclitaxel + capecitabine + trastuzumab 5 weeks, PR ▲	capecitabine + trastuzumab, 12 weeks, SD		
278	MBC	Female	50	HR+ HER2+	Prior to salvage therapy ▲	vinorelbine + platinum 4 weeks, SD ▲	vinorelbine + platinum 17 weeks, PR		
362	MBC	Female	59	HR+ HER2+	Prior to salvage therapy ▲	capecitabine + trastuzumab 36 weeks, SD			
366	NMBC	Female	62	HR+ HER2+	NMBC diagnosis, 13 days Prior to neoadjuvant therapy ▲				
400	MBC	Female	50	HR+ HER2+	Prior to salvage therapy ▲	gemcitabine + trastuzumab 7 weeks, PD ▲			
431	MBC	Female	47	HR+ HER2+	Prior to salvage therapy ▲	apatinib 8 weeks, SD	capecitabine + apatinib 6 weeks, SD		
691	NMBC	Female	56	HR+ HER2+	NMBC diagnosis, 10 days Prior to neoadjuvant therapy ▲				
694	MBC	Female	30	HR+ HER2+	Prior to salvage therapy ▲	capecitabine + pyrotinib 48 weeks, SD			
14	MBC	Female	46	HR- HER2+	Prior to salvage therapy ▲	vinorelbine + pyrotinib 5 weeks, PD ▲	vinorelbine + pyrotinib 9 weeks, SD ▲		
47	MBC	Female	57	HR- HER2+	Prior to salvage therapy ▲	capecitabine + trastuzumab + pertuzumab	capecitabine + trastuzumab + pertuzumab		

						8 weeks, PR ▲	8 weeks, PR ▲		
54	NMBC	Female	48	HR- HER2+	NMBC diagnosis, 9 days Prior to neoadjuvant therapy ▲				
73	NMBC	Female	47	HR- HER2+	NMBC diagnosis, 0 day Prior to neoadjuvant therapy ▲				
84	MBC	Female	56	HR- HER2+	MBC recurrence, 21 days Prior to salvage therapy ▲	paclitaxel + trastuzumab 6 weeks, SD ▲	paclitaxel + trastuzumab 6 weeks, SD ▲	paclitaxel + trastuzumab 13 weeks, SD	
112	MBC	Female	58	HR- HER2+	Prior to salvage therapy ▲	capecitabine 7 weeks, PD ▲	paclitaxel 6 weeks, SD ▲	paclitaxel 16 weeks, SD	
117	MBC	Female	44	HR- HER2+	Prior to salvage therapy ▲	paclitaxel + trastuzumab 12 weeks, PR	paclitaxel + trastuzumab 15 weeks, SD		
134	MBC	Female	64	HR- HER2+	Post-surgery, 54 weeks MBC recurrence, 24 days Prior to salvage therapy ▲	capecitabine + pyrotinib 51 weeks, SD			
151	MBC	Female	49	HR- HER2+	Post-surgery, 204 weeks MBC recurrence, 13 days Prior to salvage therapy ▲	paclitaxel + capecitabine + trastuzumab 23 weeks, PR ▲			
175	MBC	Female	57	HR- HER2+	Prior to salvage therapy ▲	vinorelbine + lapatinib 7 weeks, SD ▲	vinorelbine + lapatinib 16 weeks, SD		
192	MBC	Female	56	HR- HER2+	Prior to salvage therapy ▲	medroxyprogesterone + trastuzumab 5 weeks, SD ▲	medroxyprogesterone + trastuzumab 11 weeks, PD ▲	pyrotinib 11 weeks, SD ▲	
216	MBC	Female	58	HR- HER2+	Prior to salvage therapy ▲	paclitaxel + trastuzumab 17 weeks, PD ▲	pyrotinib 8 weeks, PR ▲		
274	MBC	Female	53	HR- HER2+	Prior to salvage therapy ▲	paclitaxel + trastuzumab + platinum 5 weeks, PR ▲			
289	NMBC	Female	67	HR- HER2+	NMBC diagnosis, 2 days Prior to neoadjuvant therapy ▲				
363	MBC	Female	49	HR- HER2+	Prior to salvage therapy ▲	trastuzumab	trastuzumab		

						7 weeks, PR ▲	29 weeks, SD		
368	MBC	Female	62	HR- HER2+	Prior to salvage therapy ▲	capecitabine + pyrotinib 9 weeks, PR ▲			
389	MBC	Female	58	HR- HER2+	Prior to salvage therapy ▲	capecitabine + lapatinib 6 weeks, PR ▲			
391	MBC	Female	46	HR- HER2+	MBC diagnosis, 6 days Prior to salvage therapy ▲	paclitaxel + capecitabine + trastuzumab 20 weeks, PR			
404	MBC	Female	63	HR- HER2+	Post-surgery, 163 weeks MBC recurrence, 6 days Prior to salvage therapy ▲	capecitabine + pyrotinib 7 weeks, SD ▲	capecitabine + pyrotinib 8 weeks, SD		
420	NMBC	Female	63	HR- HER2+	NMBC diagnosis, 6 days Prior to neoadjuvant therapy ▲				
433	MBC	Female	83	HR- HER2+	Post-surgery, 510 weeks MBC recurrence, 34 days Prior to salvage therapy ▲				
453	NMBC	Female	46	HR- HER2+	NMBC diagnosis, 12 days Prior to neoadjuvant therapy ▲				
587	NMBC	Female	72	HR- HER2+	NMBC diagnosis, 4 days Prior to neoadjuvant therapy ▲				
34	MBC	Female	34	TNBC	Prior to salvage therapy ▲	vinorelbine 20 weeks, PR ▲	vinorelbine 16 weeks, PD ▲		
45	MBC	Female	80	TNBC	MBC diagnosis, 6 days Prior to salvage therapy ▲	capecitabine 9 weeks, SD ▲	capecitabine 6 weeks, SD ▲	capecitabine 7 weeks, PD ▲	paclitaxel 10 weeks, PR ▲
57	MBC	Female	46	TNBC	Post-surgery, 107 weeks MBC recurrence, 6 days Prior to salvage therapy ▲	vinorelbine + platinum 7 weeks, PR ▲	vinorelbine + platinum 6 weeks, PR ▲	vinorelbine 21 weeks, PR ▲	
61	MBC	Female	51	TNBC	Post-surgery, 71 weeks MBC recurrence, 0 day Prior to salvage therapy ▲	paclitaxel 6 weeks, PD ▲			

64	MBC	Female	55	TNBC	Prior to salvage therapy ▲	capecitabine + bevacizumab 5 weeks, PD ▲			
66	MBC	Female	45	TNBC	Prior to salvage therapy ▲	capecitabine 7 weeks, SD ▲	capecitabine 7 weeks, SD ▲		
68	NMBC	Female	41	TNBC	NMBC diagnosis, 27 days Prior to neoadjuvant therapy ▲				
82	MBC	Female	60	TNBC	Prior to salvage therapy ▲	paclitaxel + toripalimab 7 weeks, SD ▲	vinorelbine + platinum 7 weeks, PR ▲		
88	MBC	Female	58	TNBC	Post-surgery, 164 weeks MBC recurrence, 6 days Prior to salvage therapy ▲	paclitaxel 10 weeks, SD ▲	paclitaxel 3 weeks, SD ▲		
93	MBC	Female	70	TNBC	Post-surgery, 63 weeks MBC recurrence, 6 days Prior to salvage therapy ▲	paclitaxel + toripalimab 10 weeks, PR ▲	paclitaxel + toripalimab 3 weeks, PR ▲	paclitaxel + toripalimab 7 week, PD	
101	MBC	Female	46	TNBC	MBC recurrence, 6 days Prior to salvage therapy ▲	paclitaxel + capecitabine 6 weeks, SD ▲	paclitaxel + capecitabine 7 weeks, SD	paclitaxel 6 weeks, PD	
131	MBC	Female	55	TNBC	Prior to salvage therapy ▲	apatinib 5 weeks, PR ▲	apatinib 10 weeks, PD		
178	MBC	Female	46	TNBC	Prior to salvage therapy ▲	paclitaxel + capecitabine 7 weeks, PR ▲	paclitaxel + capecitabine 16 weeks, PR		
226	MBC	Female	62	TNBC	Prior to salvage therapy ▲	apatinib 11 weeks, SD			
245	MBC	Female	28	TNBC	Prior to salvage therapy ▲	vinorelbine + platinum 6 weeks, PR ▲	vinorelbine + platinum 15 weeks, PR		
248	MBC	Female	31	TNBC	Prior to salvage therapy ▲	etoposide 16 weeks, PD ▲	gemcitabine 6 weeks, PD ▲		
287	MBC	Female	51	TNBC	Prior to salvage therapy ▲	paclitaxel 9 weeks, PD ▲	apatinib 7 weeks, SD ▲		
343	MBC	Female	78	TNBC	Post-surgery, 23 weeks MBC recurrence, 0 day				

					Prior to salvage therapy ▲				
384	MBC	Female	61	TNBC	Post-surgery, 450 weeks MBC recurrence, 12 day Prior to salvage therapy ▲	paclitaxel + capecitabine 7 weeks, PR ▲	paclitaxel + capecitabine 8 weeks, SD		
529	MBC	Female	37	TNBC	Post-surgery, 45 weeks MBC recurrence, 20 days Prior to salvage therapy ▲	apatinib + toripalimab 38 weeks, PD			
548	MBC	Female	43	TNBC	Post-surgery, 414 weeks MBC recurrence, 0 day Prior to salvage therapy ▲	paclitaxel + toripalimab 36 weeks, PD			
550	MBC	Female	35	TNBC	Post-surgery, 94 weeks MBC recurrence, 0 day Prior to salvage therapy ▲	bevacizumab 13 weeks, PR			
585	MBC	Female	69	TNBC	Post-surgery, 625 weeks MBC recurrence, 18 days Prior to salvage therapy ▲	paclitaxel 54 weeks, PR			

Reviewers' Comments:

Reviewer #1:

Remarks to the Author:

The revised manuscript has been improved. Authors addressed most of my comments. However, I still have a few major concerns, esp. on the rigor in their computational analyses:

(1) Cross-validation. In the revised manuscript, authors split the cohort into 100 sets of 60% training and 40% validation. Was split randomly? Also, did the authors balance the sample sizes in the training sets? Authors mentioned the leave-one-out cross-validation in the response letter for calculating weighted sums but didn't say or provide any more details whether the cross-validation was done and if so, how it addressed the overfitting, etc, especially in the revised manuscript.

(2) Weighted sums. Authors still didn't present how each marker's weight contributes to the prediction (either LDA for classification or MLR for tumor size). The low variation of ROCs as claimed does not necessarily mean that each marker's weight doesn't change. Authors need to show how each marker weights change across training sets. Actually, it is hard to tell the low variations from Fig. S13b without any quantitative measurements. Also, authors found that a single marker, EV PSMA itself can achieve a high AUC 0.8447. Does this mean that the marker dominates the prediction, probably no needing other markers?

(3) TCGA & GTEx validation. Looking at individual markers doesn't justify any potential application of their multivariate signature models (i.e., via eight markers). Authors need to validate their signature models, EVDX or EVTS from the weighted sums of eight markers using public data to see if eight markers are able to classify cancer types or predict tumor sizes. Further, how did authors remove the batches between TCGA and GTEx data?

(4) Multivariate linear regression. Using the correlation to measure the regression performance is inappropriate. Authors should report the goodness of fit like mean square errors. Moreover, Fig S4 doesn't show any linearity, even given the log x-axis.

Reviewer #2:

Remarks to the Author:

The authors have adequately addressed the comments. They performed additional experiments showing that the technology is compatible with multiple sizes, performs similar in serum and plasma; and provided additional information on marker selection and patient samples. As a minor comment I stimulate the authors to replace the terms exosomes and microvesicles with respectively small and large EVs (in agreement with MISEV2018 guidelines).

Reviewer #3:

Remarks to the Author:

The revised manuscript, more focused on detection of progression in metastatic cancers, quite thoroughly addresses the issues that this reviewer raised on the first submission. The authors add new data, new analyses, and new information that makes the manuscript much more rigorous.

Response to Reviewers

We are grateful to the reviewers for their constructive comments and insightful suggestions. Below, the reviewers' comments are shown in black, and our responses are in blue. Sections marked by red represent all altered contents in the revised manuscript. The following are our detailed responses.

Response to Reviewer #1:

Comment: The revised manuscript has been improved. Authors addressed most of my comments. However, I still have a few major concerns, esp. on the rigor in their computational analyses:

Response: We thank the reviewer for the constructive review, which has helped to improve the quality of our manuscript. The following comments have been thoroughly addressed upon the reviewer's request.

Comment 1: Cross-validation. In the revised manuscript, authors split the cohort into 100 sets of 60% training and 40% validation. Was split randomly? Also, did the authors balance the sample sizes in the training sets? Authors mentioned the leave-one-out cross-validation in the response letter for calculating weighted sums but didn't say or provide any more details whether the cross-validation was done and if so, how it addressed the overfitting, etc, especially in the revised manuscript.

Response: We thank the reviewer for this insightful comment. In the revised manuscript, the MBC monitoring cohort ($n = 78$) was randomly split into 100 sets of 60 % training ($n = 46$) and 40 % validation ($n = 32$) cohorts. As the number of PD ($n = 17$) was much smaller than that of PR/SD ($n = 61$), the sample sizes of PD and PR/SD were not balanced in the training sets. We agree with the reviewer that ROC analysis was not suitable for processing unbalanced samples. Hence, we used accuracy to evaluate the performance of classification in the training and validation sets. In the training sets, the leave-one-out cross-validation was performed to challenge the trained LDA model. The trained LDA model was then applied to the validation sets to verify its performance. To evaluate the degree of overfitting, we have compared the accuracy for PD versus PR/SD classification in all the 100 sets of training and validation cohorts. Both the training and validation sets showed high accuracies for classification, indicating little or no overfitting. The new data has been described in section "Cross-validation" of Methods and Supplementary Fig. 13.

Cross-validation. The MBC monitoring cohort ($n = 78$) was randomly split into 100

sets of 60 % training ($n = 46$) and 40 % validation ($n = 32$) cohorts. As the number of PD ($n = 17$) was much smaller than that of PR/SD ($n = 61$), the sample sizes of PD and PR/SD were not balanced in the training sets. Hence, we used accuracy to evaluate the performance of classification in the training and validation sets. In the training sets, the leave-one-out cross-validation was performed to challenge the LDA model. The trained LDA model was then applied to the validation sets to verify its performance. To evaluate the degree of overfitting, we have compared the accuracy for PD versus PR/SD classification in all the 100 sets of training and validation cohorts. Both the training and validation sets showed high accuracies for classification, indicating little or no overfitting (Supplementary Fig. 13).

Supplementary Fig. 13 | Accuracy for PD versus PR/SD classification in 100 sets training and validation cohorts. Statistical differences were determined by two-tailed, paired t test. Error bars represent the mean \pm s.d..

Comment 2: Weighted sums. Authors still didn't present how each marker's weight contributes to the prediction (either LDA for classification or MLR for tumor size). The low variation of ROCs as claimed does not necessarily mean that each marker's weight doesn't change. Authors need to show how each marker weights change across training sets. Actually, it is hard to tell the low variations from Fig. S13b without any quantitative measurements. Also, authors found that a single marker, EV PSMA itself can achieve a high AUC 0.8447. Does this mean that the marker dominates the prediction, probably no needing other markers?

Response: We thank the reviewer for this critical comment. We agree with the reviewer that the low variation of ROCs does not necessarily mean that each marker's weight doesn't change. Upon the reviewer's request, we have shown the weights of each marker across 100 training sets (Supplementary Fig. 14). Despite the existence of LDA weight variations across training sets, the mean values of each marker

weights were reflective of their different contribution to the EV^M signature. Although a single marker, EV PSMA itself, can achieve a high AUC 0.8447 (accuracy = 85.9 %), this value was still lower than that of the EV^M signature (AUC = 0.9248, accuracy = 88.5 %). In addition, we have provided the multivariate linear regression (MLR) weight of each EV marker, which varied from -57.9 for EV HER2 to 76.8 for EV CA 15-3, indicating their different contribution to the EV^{TS} signature correlated with tumor size. This data has been shown in Supplementary Figs. 14 and 15.

EV signature weights. As shown in Supplementary Fig. 14, the mean values of the LDA weights for 8 EV markers across 100 training sets varied from -0.9 to 1.8, reflecting variable contribution of different markers to the EV^M signature. Among 8 EV markers, the LDA weight was highest for EV PSMA, revealing that EV PSMA can be used to best discriminate between PD and PR/SD groups (AUC = 0.8447, 95% CI = 0.7335 – 0.9560). However, this value was still lower than that of the EV^M signature (AUC = 0.9248, accuracy = 88.5 %). In addition, we have provided the multivariate linear regression (MLR) weight of each EV marker, which varied from -57.9 for EV HER2 to 76.8 for EV CA 15-3, indicating their different contribution to the EV^{TS} signature correlated with tumor size (Supplementary Fig. 15).

Supplementary Fig. 14 | Variable contribution of different EV markers to the EV^M signature. LDA weights of 8 EV markers from 100 iterations of training sets. Error bars represent the mean \pm s.d..

Supplementary Fig. 15 | Variable contribution of different EV markers to the EV^{TS} signature. Points represent the MLR weights and error bars represent the 95% CI.

Comment 3: TCGA & GTEx validation. Looking at individual markers doesn't justify any potential application of their multivariate signature models (i.e., via eight markers). Authors need to validate their signature models, EVDX or EVTS from the weighted sums of eight markers using public data to see if eight markers are able to classify cancer types or predict tumor sizes. Further, how did authors remove the batches between TCGA and GTEx data?

Response: We thank the reviewer for this critical comment. To demonstrate the application of multivariate signature models using public data, we obtained the expression of mRNA transcripts of coding genes of the 8 protein markers for 1085 BC samples and 291 normal tissue samples from the TCGA TARGET GTEx cohort.⁶² Due to lack of tumor size information and the extremely small sample size of MBC ($n = 7$), the multivariate signature model was validated in BC versus normal classification. Using the expression data of 8 markers as inputs, the signature generated by the LDA algorithm achieved an accuracy of 95.6 % for BC versus normal classification with a PRAUC of 0.9962 (Supplementary Fig. 17). This performance was similar to that of the EV^{DX} signature (an accuracy of 96.8 % with a PRAUC of 0.9912), thus validating the combination use of 8 EV protein markers in the present study. The batches between TCGA and GTEx data have been removed in the TCGA TARGET GTEx database, because TCGA and GTEx samples are re-analyzed by the same RNA-seq pipeline and processed using a uniform bioinformatic pipeline.⁶² The new data has been described in section "TCGA assessment" of Methods and Supplementary Fig. 17.

To demonstrate the application of multivariate signature models in the TCGA TARGET GTEx database,⁶² we used the expression data of 8 markers as inputs, and the signature generated by the LDA algorithm achieved an accuracy of 95.6 % for BC versus normal classification with a PRAUC of 0.9962 (Supplementary Fig. 17). This performance was similar to that of EV^{DX} signature (an accuracy of 96.8 % with a PRAUC of 0.9912), thus validating the combination use of 8 EV protein markers in the present study.

Supplementary Fig. 17 | Assessment of expression levels of mRNA transcripts of 8 protein markers in the public database. b-c, Values (b) and PRC (c) of the signature (the weighted sum of mRNA transcript levels of the 8 markers by LDA). d, Confusion matrix showing an accuracy of 95.6 % for BC versus normal discrimination.

62 Vivian, J. *et al.* Toil enables reproducible, open source, big biomedical data analyses. *Nat. Biotechnol.* **35**, 314-316 (2017).

Comment 4: Multivariate linear regression. Using the correlation to measure the regression performance is inappropriate. Authors should report the goodness of fit like mean square errors. Moreover, Fig S4 doesn't show any linearity, even given the log x-axis.

Response: We thank the reviewer for pointing this out. In accordance with the reviewer's request, we have used both mean square errors and R squared to report the goodness of fit throughout the revised manuscript. In the original Supplementary Fig. 4, we used log x-axis to avoid the crowding of data points. To show the linearity, we now have used linear x-axis in the revised figures.

Fig. 3 | f, Concordance between tumor size and the EV^{TS} signature identified using multivariate linear regression (MLR). Mean square errors (MSE) and R squared (R^2) are indicated. Linear regression result is indicated by the dashed line.

Supplementary Fig. 3 | Correlation between tumor size and EV or plasma protein markers using linear regression. Tumor sizes measured in pretreatment MBC patients ($n = 20$) were correlated with the expression of 8 individual EV markers, SUM and plasma CA 15-3. Mean square errors (MSE) and R squared (R^2) are indicated. Linear regression was performed as indicated by the dashed line.

Supplementary Fig. 4 | Concordance between primary tumor size ($n = 11$) and the EV^{TS} signature identified using multivariate linear regression. Mean square errors (MSE) and R squared (R^2) are indicated. Linear regression result is indicated by the dashed line.

Response to Reviewer #2:

Comment: The authors have adequately addressed the comments. They performed additional experiments showing that the technology is compatible with multiple sizes, performs similar in serum and plasma; and provided additional information on marker selection and patient samples. As a minor comment I stimulate the authors to replace the terms exosomes and microvesicles with respectively small and large EVs (in agreement with MISEV2018 guidelines).

Response: We thank the reviewer for the positive evaluation of the revised manuscript. Upon the reviewer's suggestion, we have replaced the terms of exosomes and microvesicles with respectively small and large EVs throughout the revised manuscript.

Response to Reviewer #3:

Comment: The revised manuscript, more focused on detection of progression in metastatic cancers, quite thoroughly addresses the issues that this reviewer raised on the first submission. The authors add new data, new analyses, and new information that makes the manuscript much more rigorous.

Response: We thank the reviewer for the positive evaluation of our research work, and the suggestion of acceptance of this work.

Reviewers' Comments:

Reviewer #1:

Remarks to the Author:

The authors have addressed my concerns. I don't have any further comments.

Response to Reviewer #1:

Comment: The authors have addressed my concerns. I don't have any further comments:

Response: We thank the reviewer for the positive evaluation of our research work, and the suggestion of acceptance of this work.